# Type 2 Diabetes and the Multifaceted Gut-X Axes

**DOI:** 10.3390/nu17162708

**Published:** 2025-08-21

**Authors:** Hezixian Guo, Liyi Pan, Qiuyi Wu, Linhao Wang, Zongjian Huang, Jie Wang, Li Wang, Xiang Fang, Sashuang Dong, Yanhua Zhu, Zhenlin Liao

**Affiliations:** 1College of Food Science, South China Agriculture University, Guangzhou 510642, China; dawdtexlecal1983@gmail.com (H.G.); 18144779007@163.com (L.P.); 13794380338@163.com (Q.W.); wang_lin_hao@163.com (L.W.); a15362991857@163.com (Z.H.); jiewang@scau.edu.cn (J.W.); wangli_scau@scau.edu.cn (L.W.); fxiang@scau.edu.cn (X.F.); dongsashuang@126.com (S.D.); 2Department of Endocrinology & Metabolism, The Third Affiliated Hospital of Sun Yat-sen University, Guangzhou 510630, China; 3Key Laboratory of Diabetology, Guangzhou 510080, China

**Keywords:** type 2 diabetes, gut microbiome, incretin hormones, gut–liver axis, bile acids, NAFLD, short-chain fatty acids, metabolic inflammation

## Abstract

Type 2 diabetes (T2D) is a complex metabolic disease characterized by chronic hyperglycemia due to insulin resistance and inadequate insulin secretion. Beyond the classically implicated organs, emerging evidence highlights the gut as a central player in T2D pathophysiology through its interactions with metabolic organs. The gut hosts trillions of microbes and enteroendocrine cells that influence inflammation, energy homeostasis, and hormone regulation. Disruptions in gut homeostasis (dysbiosis and increased permeability) have been linked to obesity, insulin resistance, and β-cell dysfunction, suggesting multifaceted “Gut-X axes” contribute to T2D development. We aimed to comprehensively review the evidence for gut-mediated crosstalk with the pancreas, endocrine system, liver, and kidneys in T2D. Key molecular mechanisms (incretins, bile acids, short-chain fatty acids, endotoxins, etc.) were examined to construct an integrated model of how gut-derived signals modulate metabolic and inflammatory pathways across organs. We also discuss clinical implications of targeting Gut-X axes and identify knowledge gaps and future research directions. A literature search (2015–2025) was conducted in PubMed, Scopus, and Web of Science, following PRISMA guidelines (Preferred Reporting Items for Systematic Reviews). Over 150 high-impact publications (original research and review articles from *Nature*, *Cell*, *Gut*, *Diabetologia*, *Lancet Diabetes & Endocrinology*, etc.) were screened. Data on gut microbiota, enteroendocrine hormones, inflammatory mediators, and organ-specific outcomes in T2D were extracted. The GRADE framework was used informally to prioritize high-quality evidence (e.g., human trials and meta-analyses) in formulating conclusions. T2D involves perturbations in multiple Gut-X axes. This review first outlines gut homeostasis and T2D pathogenesis, then dissects each axis: **(1) Gut–Pancreas Axis:** how incretin hormones (GLP-1 and GIP) and microbial metabolites affect insulin/glucagon secretion and β-cell health; **(2) Gut–Endocrine Axis:** enteroendocrine signals (e.g., PYY and ghrelin) and neural pathways that link the gut with appetite regulation, adipose tissue, and systemic metabolism; **(3) Gut–Liver Axis:** the role of microbiota-modified bile acids (FXR/TGR5 pathways) and bacterial endotoxins in non-alcoholic fatty liver disease (NAFLD) and hepatic insulin resistance; **(4) Gut–Kidney Axis:** how gut-derived toxins and nutrient handling intersect with diabetic kidney disease and how incretin-based and SGLT2 inhibitor therapies leverage gut–kidney communication. Shared mechanisms (microbial SCFAs improving insulin sensitivity, LPS driving inflammation via TLR4, and aryl hydrocarbon receptor ligands modulating immunity) are synthesized into a unified model. An integrated understanding of Gut-X axes reveals new opportunities for treating and preventing T2D. Modulating the gut microbiome and its metabolites (through diet, pharmaceuticals, or microbiota therapies) can improve glycemic control and ameliorate complications by simultaneously influencing pancreatic islet function, hepatic metabolism, and systemic inflammation. However, translating these insights into clinical practice requires addressing gaps with robust human studies. This review provides a state-of-the-art synthesis for researchers and clinicians, underlining the gut as a nexus for multi-organ metabolic regulation in T2D and a fertile target for next-generation therapies.

## 1. Introduction

Type 2 diabetes (T2D) is a global epidemic affecting an estimated 500-million-plus individuals and placing an enormous burden on public health systems worldwide [1]. Traditionally, T2D pathophysiology was framed by dysfunction within pancreatic islets—namely, insulin-secreting β-cells and glucagon-secreting α-cells—as well as insulin-responsive tissues, such as skeletal muscle, adipose tissue, and the liver [2]. However, contemporary systems-biology work has expanded this view into a multi-organ network model that integrates crosstalk among metabolic, immune, and neuroendocrine circuits [3]. In T2D, virtually every organ system contributes to dysglycemia, as encapsulated by the “ominous octet” of mechanisms—β-cell failure, α-cell excess, hepatic- and skeletal-muscle insulin resistance, adipose dysfunction, central appetite dysregulation, heightened renal glucose reabsorption, and an impaired incretin effect [4]. Notably, the gut has emerged as a central hub that interlinks many of these pathogenic nodes through its microbiota-derived metabolites, barrier functions, and bidirectional hormonal signaling [5].

The gut communicates with metabolic organs through integrated endocrine, neural, and immune pathways that form a bidirectional “gut–organ” axis [6]. It harbors a diverse microbiota that generates bioactive metabolites and rewires host-signaling networks, thereby shaping systemic metabolism [7]. This epithelium also contains a dense enteroendocrine cell (EEC) system that secretes incretin and satiety hormones, such as GLP-1, GIP, and PYY, which modulate insulin secretion, appetite, and energy expenditure [8]. Under physiological conditions, tight junctions and mucus form a selective intestinal barrier that excludes luminal lipopolysaccharide (LPS) and other toxins from the circulation [9]. In type 2 diabetes and obesity, dysbiosis of gut microbes alongside heightened intestinal permeability (“leaky gut”) enables translocation of microbial products that provoke chronic low-grade inflammation [10]. Consequently, metabolic endotoxemia—persistently elevated circulating LPS levels—activates Toll-like receptor-4 (TLR4), an immune receptor that recognizes endotoxins, triggering NF-κB inflammation, signaling, and downstream inflammatory cascades, fostering systemic insulin resistance [11]. Concurrently, people with T2D display an impaired incretin effect in which gut-derived insulinotropic hormones lose their glucose-lowering potency [12]. Collectively, these findings position the gut as a central nexus integrating metabolic and inflammatory cues that drive diabetes pathophysiology.

Accordingly, the concept of Gut-X axes has gained prominence. These axes refer to bidirectional communication pathways between the intestine and distant organs: the gut–pancreas axis, gut–endocrine (gut–brain/adipose) axis, gut–liver axis, and gut–kidney axis. Each axis encompasses unique molecular dialogs—from nutrients and microbial metabolites to cytokines and hormones—that collectively influence glucose homeostasis and tissue health. For instance, nutrients in the gut stimulate GLP-1 release, which, in turn, amplifies insulin secretion (gut–pancreas axis). Conversely, a dysbiotic microbiome may produce excess metabolites that promote fatty liver (gut–liver axis) or uremic toxins that worsen kidney function (gut–kidney axis).

Understanding these interconnected pathways is critical because it opens new avenues for intervention. Therapies targeting the gut—e.g., probiotics, prebiotic dietary fibers, bile-acid modulators, GLP-1 receptor agonists, and even bariatric surgery—have already delivered clinically meaningful metabolic benefits [13]. For example, bariatric surgery can rapidly induce T2D remission, primarily by accelerating nutrient delivery to the distal gut, which abnormally amplifies incretin release, rather than simply reshaping gut hormone profiles or the microbiota composition [14]. Likewise, SGLT2 inhibitors lower blood glucose partly via an artificial enhancement of urinary glucose excretion—a ‘plumber-like’ effect—highlighting how harnessing an abnormal excretory route can improve glycemic control [15,16].

In this narrative review, we synthesize current evidence on how gut homeostasis interplays with the pancreas, endocrine/metabolic tissues, liver, and kidneys in the context of T2D. We first outline our literature search and appraisal methods. We then provide an overview of gut physiology and T2D pathogenesis to set the stage. The core sections detail each Gut-X axis, emphasizing mechanistic links (e.g., incretins, bile acids, short-chain fatty acids (SCFAs), LPS, and immune modulators) and summarizing key studies (Tables and Figures provided). We integrate these insights into a unified model of multi-organ crosstalk in T2D. Finally, we discuss clinical and translational implications—including how modulating the gut microbiome or gut-derived signals could complement existing T2D treatments—and highlight outstanding knowledge gaps. By examining T2D through the lens of Gut-X axes, we aim to provide researchers and clinicians with a comprehensive understanding of this emerging paradigm and inspire future investigations and therapeutic innovations.

## 2. Methods: Literature Retrieval and Appraisal

We conducted a comprehensive literature search to identify relevant articles on Gut-X axes in T2D. The search strategy followed PRISMA 2020 guidelines for transparent reporting of literature searches. We queried the electronic databases PubMed, Web of Science, and Scopus for English-language articles published between January 2015 and April 2025. We followed a structured search strategy, but our review is narrative, with an emphasis on seminal studies, regardless of the date. We explicitly note that older foundational work was considered; despite the initial 2015 cutoff date, older seminal works (pre-2015) are cited occasionally to provide historical context (e.g., discovery of incretins and initial microbiome–metabolism links). Core search terms included combinations of “type 2 diabetes” AND (“gut microbiota” OR “microbiome” OR “intestinal”) AND (“pancreas” OR “incretin” OR “GLP-1” OR “liver” OR “NAFLD” OR “bile acids” OR “kidney” OR “uremic toxin” OR “endocrine” OR “hormone” OR “axis”). Additional keywords, like “short-chain fatty acid”, “lipopolysaccharide”, and “inflammation”, and specific organ terms were used to refine the results. Reference lists of pertinent review articles were also screened for any studies missed in the database search.

Our search initially yielded ~1200 articles. After removing duplicates, two reviewers (authors) independently screened titles and abstracts for relevance. We included original research (clinical trials, cohort studies, and translational experiments) and reviews that provided insight into mechanistic links between the gut and metabolic organs in the context of T2D or related metabolic diseases. We excluded studies focusing solely on type 1 diabetes or other forms of diabetes and those where gut variables were not a central aspect. In total, about 200 articles were deemed relevant and retrieved for full-text evaluation.

Each included study was appraised for methodological quality and strength of evidence. We then prioritized high-tier sources and certainty ratings in line with the updated GRADE guidance [17]. For example, human interventional studies (e.g., randomized controlled trials of probiotics or dietary interventions) and large-scale observational studies were weighted strongly for clinical insights, whereas mechanistic animal studies were used to elucidate pathways not easily studied in humans. We also gave preference to publications in high-impact journals (*Nature*, *Cell*, *Lancet*, *Gut*, *Diabetologia*, etc.) and to more recent evidence (e.g., post 2015) to ensure up-to-date conclusions. However, older seminal works (pre 2015) are cited occasionally to provide historical context (e.g., discovery of incretins and initial microbiome–metabolism links).

## 3. Gut Homeostasis and T2D Pathophysiology: An Overview

Gut Homeostasis: The gastrointestinal tract is not only crucial for nutrient digestion and absorption but also serves as a central hub for metabolic and immune homeostasis [18]. A healthy gut maintains a harmonious microbial community, an intact mucosal barrier, and a balanced immune response [18]. The adult human gut microbiota is dominated by the bacterial phyla Firmicutes and Bacteroidetes, which hundreds of constituent species collaborate in fiber fermentation, vitamin synthesis, and pathogen defense [19]. These commensal microbes generate many bioactive metabolites—most prominently, the short-chain fatty acids (SCFAs) acetate, propionate, and butyrate—that act as both energy substrates and host-signaling molecules [20]. SCFAs bind to G-protein-coupled receptors, such as GPR41/FFAR3 and GPR43/FFAR2, on enteroendocrine and immune cells, tuning hormone secretion and inflammatory tone [20]. Concurrently, the intestinal epithelium forms a tight-junction-regulated barrier that allows selective nutrient uptake while blocking translocation of microbial components (e.g., LPS and peptidoglycans) into the circulation [9]. Gut equilibrium is further reinforced by secretory IgA and antimicrobial peptides that restrain microbial overgrowth and preserve ecological balance [19].

**T2D Pathophysiology:** T2D arises from a combination of insulin resistance in peripheral tissues and β-cell dysfunction in the pancreas [21]. In the pre-diabetic state, the skeletal muscle and liver become progressively less responsive to insulin, leading to reduced glucose uptake and increased hepatic glucose output [22]. Pancreatic β-cells initially compensate by hypersecreting insulin, but chronic glucotoxic and lipotoxic stresses drive β-cell dedifferentiation and apoptotic loss [23]. By the time T2D is diagnosed, β-cell function is typically diminished by ~50% relative to that in healthy individuals [24]. Moreover, inappropriate hypersecretion of glucagon from pancreatic α-cells further aggravates hyperglycemia [25]. Systemic low-grade inflammation accompanying obesity and T2D—marked by cytokines such as TNF-α and IL-6—impairs insulin signaling and recruits immune cells into insulin-sensitive tissues [26].

Crucially, the gut contributes to each of these facets. First, the phenomenon of incretin dysfunction in T2D links the gut and pancreas: In healthy individuals, oral glucose elicits ~2–3 times more insulin than intravenous glucose due to gut hormones (the incretin effect), whereas T2D patients show an attenuated incretin effect [8]. Specifically, glucose-dependent insulinotropic polypeptide (GIP) action is blunted in T2D, and while GLP-1 action is relatively preserved, overall incretin responsiveness is impaired, despite generally preserved hormone secretion [27]. This gut hormonal defect contributes to insufficient postprandial insulin release [8]. Second, gut microbial dysbiosis and increased intestinal permeability in obesity/T2D have been associated with metabolic endotoxemia, which is proposed to contribute to insulin resistance [28]. A high-fat diet in mice raises plasma LPS levels and triggers weight gain, inflammation, and insulin resistance (a concept now supported in humans as well) [29]. T2D patients have been found to exhibit higher circulating LPS levels and an activated TLR4 pathway in adipose and muscle tissues, correlating with insulin-resistance severity [30]. Thus, a leaky gut can broadcast inflammatory signals systemically, impairing insulin action and possibly β-cell health [28]. Third, certain gut microbiota compositions may influence nutrient metabolism in ways that promote diabetes [31]. For example, an enrichment of branched-chain amino-acid (BCAA)-producing gut bacteria (such as *Prevotella copri* and *Bacteroides vulgatus*) was associated with elevated BCAAs and insulin resistance in humans [32]. BCAAs are known to activate mTOR and inflammatory pathways in muscle, contributing to insulin resistance; thus, microbiota that increase BCAA biosynthesis could worsen metabolic health [33]. Conversely, a loss of beneficial SCFA-producing bacteria (e.g., *Faecalibacterium prausnitzii* and *Roseburia*) is often noted in T2D and metabolic syndrome, potentially reducing SCFA-mediated benefits in insulin sensitivity and gut integrity [34].

In summary, gut dysregulation (whether through microbiome alterations, impaired incretin hormone release, or increased gut permeability) is now recognized as a key upstream event in the cascade leading to T2D. The graphical abstract encapsulates how an unhealthy diet and genetic predisposition might initiate gut microbial shifts and barrier dysfunction, unleashing a chain reaction of hormonal imbalance and inflammation affecting multiple organs. The following sections delve into each specific Gut-X axis, detailing how these pathways operate and interact. Improved clarity on these mechanisms is essential for devising interventions that restore gut homeostasis and thereby ameliorate metabolic disturbances in T2D.

## 4. Gut–Pancreas Axis

The gut–pancreas axis refers to the bidirectional communication between the intestine and the pancreatic islets that regulates glucose homeostasis. The most well-characterized components of this axis are the incretin hormones produced by enteroendocrine cells, which markedly enhance postprandial insulin secretion. In addition, gut-derived metabolites and neural signals can influence pancreatic β-cell function and survival. Figure 1 schematically illustrates the gut–pancreas axis, and Table 1 summarizes selected studies highlighting gut–pancreatic interactions in T2D.

**Incretin Physiology and T2D:** Incretins are hormones, released from the gut in response to nutrient ingestion, that potentiate glucose-stimulated insulin secretion from β-cells [8]. The two primary incretins are GLP-1 (glucagon-like peptide-1), secreted by L-cells in the ileum and colon (though colon-derived GLP-1 does not meaningfully enter circulation—e.g., total-colectomy patients have normal GLP-1 levels), and GIP (glucose-dependent insulinotropic polypeptide), secreted by K-cells in the duodenum and jejunum [8,35]. Together, GLP-1 and GIP account for ~50–70% of the insulin response to oral glucose loads in healthy individuals [36]. GLP-1 also suppresses glucagon from α-cells and slows gastric emptying, helping to limit postprandial glycemic excursions [35]. In T2D, as noted, the incretin effect is attenuated—a defect that precedes clinical diabetes [8]. Patients with T2D often have near-normal or slightly reduced GLP-1 levels but impaired GIP action [8,35]. Nonetheless, the preserved responsiveness to GLP-1 in T2D laid the groundwork for incretin-based therapies [37]. Exogenous GLP-1 receptor agonists (e.g., exenatide, liraglutide, and semaglutide) can still robustly stimulate insulin and lower glucose levels in T2D [37]. These agents have become pillars of T2D treatment, underscoring the gut–pancreas axis’s importance. Notably, GLP-1 RAs require sufficient residual β-cell function to work—they stimulate insulin only if pancreatic β-cells are present and responsive [37].

Mechanistically, nutrient sensing in the gut triggers incretin release via complex pathways [38]. Oral glucose stimulates GIP release from K-cells in the upper intestine, while fats and carbohydrates reaching the distal ileum stimulate GLP-1 release from L-cells [8,15]. Bile acids can activate TGR5 on L-cells to induce GLP-1 secretion. Non-nutritive sweeteners can engage L-cell sweet-taste receptors (T1R2/T1R3), but rigorous human studies show no significant GLP-1 release effect; evidence for sweeteners stimulating GLP-1 is conflicting [39,40]. Once released, incretins travel through the portal circulation to the pancreas. GIP acts directly on β-cell GIP receptors to augment insulin release (in a glucose-dependent manner), and GLP-1 acts on GLP-1 receptors on β-cells, as well as indirectly via vagal afferents in some cases [8,41]. In T2D, although GIP levels may be normal or even elevated, β-cells become refractory to GIP’s insulinotropic effect, whereas GLP-1 retains partial activity—a selective “incretin resistance” for GIP that remains incompletely understood [8]. Interestingly, tirzepatide’s remarkable efficacy was somewhat unexpected: It was not simply the sum of GLP-1 + GIP effects (GIP/GLP-1 co-agonists did not outperform GLP-1 alone), suggesting tirzepatide’s benefits stem from unique features beyond dual agonism. Its discovery was essentially serendipitous, reflecting optimized properties rather than just GIP addition [42]. This progress has revitalized research into the gut–pancreas axis, expanding the “incretin universe” to potential co-agonists and even tri-agonists (GIP/GLP-1/glucagon), such as the triple agonist retatrutide now advancing in clinical trials [43]. By contrast, combining GLP-1 with glucagon agonism shows clearer synergy—GLP-1 + glucagon dual agonists confer distinct benefits (glucagon can activate GLP-1 receptors and increase energy expenditure), suggesting a more effective complementation than GIP provides.

**Microbiota and β-Cell Function:** Beyond hormones, gut microbes and their metabolites are increasingly recognized as modulators of islet function—a newer dimension of the gut–pancreas axis [44]. Changes in the gut microbiome can alter levels of circulating nutrients and signaling molecules that affect β-cells [44]. One clear example involves short-chain fatty acids (SCFAs) produced by fiber-fermenting bacteria [45]. SCFAs, especially butyrate and propionate, can stimulate GLP-1 secretion from L-cells by activating FFAR2/3 receptors, leading to augmented incretin-mediated insulin release [46]. Butyrate also has anti-inflammatory effects that may indirectly benefit β-cell health by reducing systemic inflammation [47]. Propionate, on the other hand, serves as a gluconeogenic substrate in the liver—so its net metabolic effects are complex—but moderate propionate production is linked with satiety signaling via gut–brain pathways [48]. Diets high in fermentable fiber enrich SCFA-producing bacteria and have been shown to improve glycemic control in T2D [49]. A clinical trial provided T2D patients with a diversified high-fiber diet and observed increased butyrate-producing gut bacteria, elevated fasting GLP-1 levels, and significant reductions in HbA1c levels compared to those in patients with a control diet [50]. This underscores how manipulating the gut microbiota can enhance gut–pancreatic endocrine signaling and glycemic outcomes in humans [50].

Another link is through amino acid metabolism [51]. Certain gut bacteria metabolize dietary amino acids and in doing so, influence the availability of amino acids that are crucial for islet function. Tryptophan, for instance, can be metabolized by gut microbes to indole derivatives that activate aryl hydrocarbon receptors (AhRs)—some of these metabolites (like indole-3-propionic acid, IPA) appear to be beneficial [51,52]. Higher circulating levels of IPA, a microbial tryptophan metabolite, were associated with a lower risk of T2D and with better β-cell function in the Finnish Diabetes Prevention Study [53]. Although IPA’s mechanism is not fully proven, it may act as an antioxidant and an anti-inflammatory agent, protecting β-cells from oxidative stress [52]. Conversely, gut microbes that produce excessive branched-chain amino acids (BCAAs) can negatively impact β-cells [54]. Elevated BCAA levels are linked to not only insulin resistance but also β-cell overload (as BCAAs can overstimulate insulin secretion acutely and impair it chronically) [54]. Colonizing mice with Prevotella copri (a BCAA-producing bacterium associated with insulin resistance in humans) led to worsened glucose tolerance and higher circulating BCAA levels [55,56]. This suggests a causal role of microbiota in modulating nutrient signals that reach the pancreas.

Inflammation is another route: Gut permeability, leading to LPS translocation, can cause systemic inflammation that injures β-cells [57]. In type 2 diabetes, a ‘leaky’ gut allows persistent translocation of intestinal antigens (e.g., LPS and peptidoglycan) into the circulation, leading to chronic stimulation of innate immune receptors. TLR4 dysfunction—such as upregulated receptor expression and impaired feedback regulation—means that even low-level endotoxin exposure continually triggers inflammatory cascades. This persistent TLR4 activation contributes to prolonged low-grade inflammation, insulin resistance, and β-cell injury in T2D. Chronic exposure of islets to low-dose LPS has been shown to induce NLRP3-inflammasome activation in β-cells and islet macrophages, promoting β-cell dedifferentiation and death [58]. A recent study found that trimethylamine-N-oxide (TMAO), a metabolite produced by gut microbes from dietary choline/carnitine, directly impairs insulin secretion and triggers β-cell stress via NLRP3 activation [59]. TMAO levels are elevated in T2D and predictive of future diabetes [60], and markedly suppressing its formation through genetic or pharmacological inhibition of the hepatic FMO3 enzyme has been proposed to improve insulin secretion and glycemic control in diabetic mice [59]. These findings connect a gut-derived metabolite to pancreatic β-cell dysfunction, reinforcing the pathological gut–pancreas link.

**Neural Gut–Islet Signals:** Although hormonal and metabolic signals dominate, neural pathways also participate in the gut–pancreas axis. Ingestion of food activates vagal afferents in the gut, which relay to the brainstem and then, via efferents, can trigger a cephalic phase of insulin secretion, even before the glucose level rises [61]. This neurogenic early insulin response is partly mediated by cholinergic innervation of islets and can be influenced by gut factors. For example, vagal stimulation by nutrients or gut hormones, such as GLP-1 (which can act on vagal receptors), primes insulin secretion [62]. Additionally, the sympathetic nervous system—invariably upregulated in metabolic syndrome—can inhibit insulin secretion via adrenergic receptors on β-cells [63]. Gut dysbiosis has been linked to altered activity of the gut–brain axis; for instance, microbial metabolites can reach the brain and modify vagal tone [64]. Though less studied than incretins, these neural routes provide further insight into how the gut environment might acutely modulate islet function.

**Evidence in Humans:** Clinically, several lines of evidence underscore the importance of the gut–pancreas axis in humans with T2D. Perhaps the most striking comes from bariatric surgery. Procedures like gastric bypass markedly improve β-cell function and glycemic control within days—faster than can be explained by weight loss—owing partly to heightened GLP-1 release and altered gut nutrient flow (the “hindgut hypothesis”) [65]. Surgical patients often experience an exaggerated incretin response after meals, contributing to diabetes remission in ~50–80% of cases (depending on the baseline severity and diabetes duration) [65]. Another piece of evidence is fecal microbiota transplantation (FMT). In a randomized trial, transferring stool from lean, healthy donors to males with metabolic syndrome produced a significant improvement in peripheral insulin sensitivity after 6 weeks [66]. The benefit correlated with enrichment of butyrate-producing gut bacteria in recipients. Although the effect waned by 18 weeks (highlighting the need for sustained dietary change), the study demonstrated that modulating the microbiome can directly influence glycemic control and, by extension, β-cell–insulin axis function in humans. Additionally, probiotic supplementation has produced modest but significant improvements in insulin secretion and HbA1c in several trials. A 2023 meta-analysis of 30 RCTs involving 1827 participants reported that probiotics reduced HbA1c by ~0.2% and improved HOMA-IR, indicating better insulin sensitivity and possible β-cell relief [13]. Certain strains, such as *Akkermansia muciniphila*, are now under investigation for T2D therapy; a recent 12-week, randomized, double-blind trial in overweight/obese T2D patients showed that pasteurized *A. muciniphila* improved insulin sensitivity and modestly lowered fasting insulin levels, with efficacy depending on the baseline microbial abundance [67]. Intriguingly, a secreted protein (P9) from *A. muciniphila* has been shown to stimulate GLP-1 release from intestinal L-cells, offering a mechanistic link whereby augmenting this microbe could boost incretin output and, consequently, insulin secretion [68].

## 5. Gut–Endocrine Axis (Gut–Brain and Gut–Adipose Crosstalk)

The “gut–endocrine axis” refers broadly to the gut’s communication with the body’s endocrine and neuroendocrine systems beyond the pancreas [69,70]. This encompasses gut–brain signaling (which influences appetite, satiety, and energy expenditure) as well as gut interactions with adipose tissue and other hormone-secreting organs [70,71,72]. Essentially, it is the hormonal crosstalk orchestrated by enteroendocrine cells and neural circuits that integrate nutrient information from the gut with whole-body metabolic regulation [69,71]. Dysregulation of this axis can contribute to obesity and T2D by perturbing hunger/satiety signals and adipokine profiles [70,72]. Restoring the gut–endocrine balance—for instance via bariatric surgery or pharmacotherapy—has proven to be effective in diabetes remission, highlighting its importance [73,74]. Figure 2 schematically illustrates the gut–endocrine axis.

**Gut–Brain Axis and Appetite Regulation:** The gut and brain are tightly linked via neural (vagal and spinal afferents) and endocrine pathways [75]. After a meal, the intestines secrete several hormones that act on the brain’s appetite centers (in the hypothalamus and brainstem) to induce satiety [75,76]. Key among these are GLP-1 and Peptide YY (PYY) from L-cells and oxyntomodulin (a GLP-1-related peptide) from the gut [76,77]. We clarify that peripheral GLP-1 does not readily cross the blood–brain barrier. Instead, it acts via areas lacking a full BBB (e.g., the area postrema in the brainstem). GLP-1 receptors are highly expressed in such circumventricular organs, allowing GLP-1 to influence the brain by interacting with neurons there [76,77]. PYY, co-secreted with GLP-1 postprandially, acts on Y2 receptors in the hypothalamus to suppress hunger [78,79]. In parallel, the stomach produces ghrelin during fasting—a hunger hormone, which is paradoxically low in individuals with obesity, that increases appetite by acting on the hypothalamus (ghrelin levels normally drop after eating) [80]. In obesity and T2D, there is often resistance to satiety signals and possibly blunted postprandial PYY or GLP-1 responses, contributing to hyperphagia [78]. Some T2D patients have elevated fasting ghrelin levels (in those who lose weight, ghrelin levels rise, which makes maintenance difficult) [80,81]. Thus, an imbalanced gut–brain axis can perpetuate excess caloric intake and weight gain, worsening insulin resistance [75,76,80].

Bariatric surgery provides dramatic evidence of gut–brain axis modulation [82]. Roux-en-Y gastric bypass (RYGB) is not actually “restrictive”—it rapidly funnels nutrients to the distal small intestine, and the reduced food intake is due almost entirely to diminished appetite and early satiety (from high GLP-1/PYY and low ghrelin levels) rather than mechanical restriction [83,84]. Many patients report early satiety and decreased preferences for high-calorie foods after RYGB—effects attributable to these hormonal changes [85]. T2D remission rates after RYGB or sleeve gastrectomy (which also increases GLP-1 levels) are high (60–80% at 1 year) [86]. One hypothesis (often termed the “hindgut hypothesis”, though it actually involves the mid-gut) posits that expedited nutrient delivery to the ileum increases L-cell stimulation (GLP-1, PYY), while the “foregut hypothesis” suggests exclusion of the duodenum alters unknown signals [14]. Regardless, the success of metabolic surgery underscores that harnessing the gut–brain–endocrine axis can powerfully improve glucose metabolism independent of weight loss (some effects occur within days) [87].

Pharmacologically, the gut–brain axis is targeted by incretin-based therapies and emerging polypeptide hormone co-agonists. GLP-1 receptor agonists not only enhance insulin secretion but also act centrally to curb appetite, producing clinically meaningful weight losses of about 8–15% [88]. This weight reduction, in turn, lowers insulin resistance and improves glycemic control in T2D [88]. Some combined GIP/GLP-1 effects have been observed (demonstrated in rodents; human relevance is unknown). The dual GIP/GLP-1 agonist tirzepatide achieves still greater effects, driving ~15–20% weight loss together with large HbA1c declines that frequently return values to the normoglycemic range [42,89]. Intriguingly, although GIP alone was once considered as orexigenic, in co-agonism with GLP-1, it augments weight loss; mechanistic studies suggest GIP may amplify GLP-1 signaling in key hypothalamic and brain-stem circuits or exert central effects only at supra-physiological concentrations [62]. Amylin, another gut-related hormone (co-secreted with insulin), also suppresses appetite and glucagon; the amylin analog pramlintide is approved for diabetes, and longer-acting amylin or amylin-plus-peptide co-agonists are advancing in obesity pipelines [90]. Success with such multi-hormone agents—including triple GLP-1 + GIP + glucagon agonists, like retatrutide—illustrates a “medical bariatric surgery” concept that recreates the post-surgical endocrine milieu to drive weight loss and potent glucose lowering [91,92].

**Gut–Adipose Axis:** The gut also communicates with adipose tissue in ways that influence the release of adipokines (such as leptin and adiponectin) and systemic lipid metabolism [72,93]. One link is through diet-driven changes in bile acids: High-fat diets that remodel the gut microbiota can generate bacterial metabolites that reach adipose tissue or the liver, modulating lipogenesis and fat storage [94,95]. Chronic inflammation originating in the gut—exemplified by lipopolysaccharide (LPS) leakage—can infiltrate adipose depots via TLR4-dependent activation of resident macrophages, triggering local cytokine production that induces adipose insulin resistance and disrupts adipokine secretion, which have been suggested to promote adipose browning and to activate the TLR4–NF-κB pathway in adipose macrophages, but robust human evidence for these effects is lacking [95,96]. This creates a vicious cycle: Inflamed adipose tissue secretes less adiponectin (an insulin-sensitizing hormone) and more resistin and proinflammatory cytokines, thereby exacerbating systemic insulin resistance [93].

There is also evidence that gut hormones influence adipose tissue directly or indirectly [97]. GLP-1 receptors on adipose tissue have been identified in several studies, and GLP-1 may promote lipolysis or the browning of adipose tissue—findings are mixed, but multiple reports suggest that GLP-1 analogs increase energy expenditure [97,98]. Gut microbes generate short-chain fatty acids (SCFAs) that bind to receptors on adipocytes, such as GPR43 [99,100]. Activation of GPR43 by propionate or butyrate in adipose tissue has been shown to inhibit insulin signaling in adipocytes yet paradoxically limit fat accumulation—mouse studies indicate that SCFAs acting through GPR43 provide negative feedback on adipose expansion, diverting energy to other tissues and enhancing systemic insulin sensitivity [99,100,101]. In humans, circulating SCFAs correlate with specific adipokines and gut peptides, although these relationships are still being elucidated [101].

**Endocrine Organ Crosstalk:** It is worth noting that the “endocrine axis” can also imply gut interactions with other endocrine organs (thyroid, adrenal gland, etc.) [102]. For instance, gut microbiota may modulate cortisol metabolism by affecting the conversion of cortisone to cortisol in the liver and adipose tissue (via microbial bile acid and steroid metabolism) [103]. Some studies have found that probiotics can lower urinary free cortisol levels, suggesting a link between gut microbes and the hypothalamic–pituitary–adrenal (HPA) axis [104]. Chronic stress and elevated glucocorticoid levels promote visceral fat and hyperglycemia, thus a gut–HPA connection might influence diabetes risk [105]. The gut–thyroid axis is another emerging concept: Gut bacteria can deconjugate thyroid hormones and influence iodine uptake, and hypothyroidism can slow gut motility, affecting the microbiota [102]. However, direct links to T2D are less clear here.

**Illustrative Example—PCOS:** An example of gut–endocrine interplay is polycystic ovary syndrome (PCOS), an endocrine–metabolic disorder often accompanied by insulin resistance [106]. Studies have found gut-microbiota dysbiosis in PCOS, and treating affected women with probiotics, prebiotics or synbiotics improves not only metabolic parameters but also sex-hormone profiles [107,108]. This suggests that gut microbes can modulate the systemic endocrine milieu—perhaps by altering insulin dynamics and thereby ovarian androgen production. While PCOS itself lies outside our main scope, it clearly exemplifies how gut-directed interventions can exert far-reaching endocrine effects.

**Clinical Perspective:** For T2D patients, addressing the gut–endocrine axis means tackling appetite and weight, which is fundamental for long-term diabetes control. Lifestyle interventions that improve gut hormone responses—high-protein diets (which strongly stimulate PYY and GLP-1) or time-restricted feeding (to align with circadian gut hormone rhythms)—have shown promise in improving glycemia and weight [109,110]. There is also interest in neuromodulation techniques, e.g., vagus nerve stimulation to mimic the afferent signals of satiety or gut-targeted nutrient formulations that preferentially stimulate L-cells [111,112]. An example is “preload” shakes rich in certain amino acids or bitter tastants that can amplify GLP-1/PYY before a meal and reduce subsequent caloric intake (a strategy tested in obesity clinics) [112,113].

## 6. Gut–Liver Axis

The gut–liver axis is a well-recognized bidirectional pathway due to the anatomical connection via the portal vein [114]. Blood from the intestines drains directly into the liver, carrying not only nutrients but also microbial metabolites, for instance, certain microbiota-derived molecules, which activate host receptors, such as the aryl hydrocarbon receptor (AhR), influencing gut immune responses, and endotoxins [115]. The liver, in turn, secretes bile acids into the gut, which shape the microbiome [116]. This intimate connection means that gut dysbiosis or barrier dysfunction can have immediate effects on hepatic metabolism and inflammation—a crucial factor in conditions like non-alcoholic fatty liver disease (NAFLD), also recently termed as metabolic-dysfunction–associated fatty liver disease (MAFLD), which commonly coexists with T2D [117]. Conversely, liver-derived factors (bile acids, IgA, etc.) influence the gut microbial environment [118]. In T2D, the gut–liver axis often becomes perturbed, contributing to insulin resistance, hepatic steatosis, and systemic inflammation [119]. Figure 3 schematically illustrates the gut–liver axis, and Table 2 summarizes selected studies highlighting gut–liver interactions in T2D.

**Bile Acids as Signaling Molecules:** Bile acids (BAs) are synthesized from cholesterol in the liver, stored in the gallbladder, and released into the intestine to aid fat digestion [120]. Importantly, BAs also act as hormonal regulators by activating receptors such as FXR (the farnesoid X Receptor) and TGR5 (the G-protein-coupled bile acid receptor) [121,122]. The gut microbiota modifies bile acids extensively: Bacteria deconjugate primary BAs (cholic and chenodeoxycholic acids) and convert them to secondary BAs (deoxycholic acid, lithocholic acid, etc.) [123]. This microbial biotransformation modulates which BAs circulate and which receptors are activated [123].

Activation of the FXR in the ileum by bile acids induces the hormone FGF19 (in humans; Fgf15 in mice), which travels to the liver to suppress bile acid synthesis and, partly, gluconeogenesis [124]. In the liver, FXR activation improves insulin sensitivity and reduces triglyceride synthesis [125]. In T2D and NAFLD, FXR signaling can be dysregulated; for instance, circulating FGF19 concentrations are often lower in patients with NAFLD, permitting unchecked bile acid synthesis and potential accumulation of toxic species [126]. TGR5, expressed on intestinal L-cells and Kupffer cells (liver macrophages), when activated by certain bile acids, such as lithocholic acid and deoxycholic acid, stimulates GLP-1 secretion from L-cells and dampens macrophage inflammation through cAMP-dependent pathways [127,128]. Thus, bile acids act as gut–liver axis hormones that influence glucose homeostasis both by potentiating incretin release and by modulating hepatic glucose and lipid metabolism.

Dysbiosis can alter the bile acid pool in ways that worsen metabolism [129]. For example, a microbiome with more 7α-dehydroxylating bacteria (which produce secondary BAs, like DCA) might lead to higher circulating DCA levels [130]. DCA has been implicated in liver insulin resistance and even hepatocellular carcinoma promotion by causing hepatic stellate cellular senescence [130,131]. On the other hand, a microbiome favoring ursodeoxycholic acid (UDCA) production or retention might be metabolically protective (UDCA is a secondary BA with less FXR activation, often considered as hepatoprotective) [132]. Treatment with obeticholic acid (an FXR agonist) has been tested in NASH (a severe form of NAFLD)—it did improve liver histology in some patients but with side effects [133]. Its effects on glycemia were mixed, but it highlights targeting FXR as a gut–liver axis therapeutic strategy [133].

**NAFLD and Metabolic Syndrome:** NAFLD is present in the majority of T2D patients [134]. It is characterized by hepatic fat accumulation and can progress to inflammation (NASH) and fibrosis [134]. The gut–liver axis is central to NAFLD pathogenesis [135]. Gut-derived LPS, as discussed, can enter the portal circulation when the intestinal barrier is compromised, directly activating Kupffer cells (resident macrophages) in the liver via TLR4 [135,136]. This provokes the release of proinflammatory mediators (TNFα and IL-1β) that promote liver insulin resistance and hepatocyte injury [136]. Animal models have shown that high-fat diets cause both dysbiosis and a leaky gut, leading to elevated portal LPS and the onset of steatohepatitis; preventing this via antibiotics or TLR4 knockout protects against NAFLD [137,138]. In humans, evidence includes findings of higher plasma LPS levels and endotoxin-related markers in NAFLD patients, correlating with disease severity [135]. Furthermore, small intestinal bacterial overgrowth (SIBO) is more prevalent in NAFLD and is associated with increased gut permeability and NASH severity, suggesting the small-bowel microbiome might also contribute [139].

Besides LPS, other microbial metabolites link to NAFLD. One is ethanol: Certain gut bacteria (e.g., Escherichia coli strains) ferment carbs to ethanol. Patients with NASH have been found to have higher blood ethanol levels and more ethanol-producing bacteria in their gut than weight-matched controls [140]. Chronic endogenous ethanol production could contribute to fatty liver, akin to alcoholic liver disease, by driving oxidative stress [140]. Another metabolite, trimethylamine (TMA), from gut microbes (and originating from dietary choline and carnitine), is converted in the liver to TMAO. While TMAO is more notorious in cardiovascular disease, some studies suggest it may also worsen NAFLD by altering cholesterol metabolism or inducing inflammation in the liver [141,142]. Choline itself is essential for VLDL export in the liver; gut microbiota that deplete choline (by converting it to TMA) can induce fatty liver owing to choline deficiency [143]. Indeed, mice fed a choline-deficient diet develop fatty liver unless their microbiome is eradicated, implicating microbial choline metabolism [137].

**Insulin Resistance and Hepatic Gluconeogenesis:** The liver is a major site of insulin’s action (to suppress glucose production). Hepatic insulin resistance is largely driven by the accumulation of fat in the liver—hepatic steatosis is a primary cause of insulin resistance in obesity and T2D (and is extremely common in these conditions). In T2D, insulin fails to suppress hepatic gluconeogenesis. Excessive glucagon (hyperglucagonemia) is a major contributor to elevated hepatic glucose output—inappropriate α-cell secretion overstimulates hepatic gluconeogenesis. (Therapeutically, targeting glucagon signaling is therefore an important strategy in controlling hepatic metabolism.)—causing high fasting glucose levels. Gut-derived factors can influence this. As mentioned, FGF19 (from intestinal FXR activation) normally helps to suppress gluconeogenesis after meals [124]. Some T2D patients have lower postprandial FGF19 levels, which might lead to excessive hepatic glucose output [144]. Conversely, butyrate and propionate from the gut can travel via the portal vein to the liver; butyrate is mostly used by colonocytes, but some reaches the liver and has been shown to improve insulin signaling [145], whereas propionate is a substrate for gluconeogenesis. There is a debate about whether chronic high propionate (e.g., as a food preservative) levels might raise gluconeogenesis and insulin resistance [146], but physiologic SCFAs generated from dietary fiber are largely beneficial [147]. Another link is chronic inflammation: Cytokines such as TNFα (possibly originating from gut-LPS-activated macrophages) induce hepatic insulin resistance by inhibiting insulin-receptor signaling [26]. Therefore, a proinflammatory gut–liver axis will exacerbate the core defects of T2D.

**Therapeutic/Translational Aspects:** Targeting the gut–liver axis is a promising approach for both NAFLD and T2D. Some strategies and evidence include:

**Probiotics/Synbiotics:** Several RCTs in NAFLD have tested probiotics. Meta-analyses indicate that such interventions modestly reduce hepatic aminotransferases and intrahepatic fat [148]. For example, a 24-week randomized controlled trial of a multi-strain probiotic in biopsy-proven NASH (the PROBILIVER study) improved insulin resistance and lowered AST levels [149]. These benefits are thought to arise from reinforcement of the intestinal barrier—resulting in lower portal LPS—and the production of beneficial metabolites, such as short-chain fatty acids [117]. While not curative, probiotics, therefore, show promise as adjunct therapies that modulate the gut–liver axis.

**Dietary Fiber and Fecal Transplants:** As in T2D, high-soluble-fiber diets appear to benefit NAFLD by boosting colonic SCFA production and tightening the gut barrier, thereby lowering portal endotoxin influx [150,151]. One pilot randomized study showed that fecal microbiota transplantation from a lean donor to obese NAFLD patients improved hepatic insulin sensitivity and reduced the liver-fat fraction at 6 months—effects that were the greatest when the donor’s stool harbored high microbial diversity [152]. This underscores the microbiome’s causal role in hepatic fat regulation.

**Bile-Acid-based therapies:** Obeticholic acid (an FXR agonist) showed histological improvement of fibrosis in NASH in the phase 3 REGENERATE trial, but it also raised levels of LDL-cholesterol—a known on-target effect of hepatic FXR activation [153]. Other approaches include bile-acid sequestrants, such as colesevelam: Originally licensed for hypercholesterolemia, colesevelam also lowers HbA1c levels by ~0.5 percentage points in type 2 diabetes, as demonstrated in the GOAL-RCT [154]—likely by interrupting enterohepatic cycling (forcing de novo bile-acid synthesis, which consumes hepatic glucose) and/or by enhancing GLP-1 release through TGR5 activation. Indeed, bile-acid-induced TGR5 signaling on intestinal L-cells has been shown to amplify postprandial GLP-1 secretion and improve glucose tolerance [127]. Thus, pharmacologically manipulating bile-acid pathways can beneficially affect both liver histology and glycemic control.

**TLR4 Inhibition:** Given that LPS–TLR4 signaling drives hepatic inflammation, TLR4 antagonists have been explored as therapeutics [155]. A small-molecule antagonist, JKB-121 (a TLR4 antagonist), has not shown significant efficacy in clinical trials—a phase II NASH study found no improvement versus a placebo. (Despite promising preclinical rationale, JKB-121 failed to demonstrate benefits in human NASH patients). Although the study did not significantly improve liver fat or fibrosis biomarkers, it illustrates efforts to block gut-derived endotoxin signaling [156,157]. Lifestyle remains pivotal: weight-loss programs based on diet and exercise lower intestinal permeability and circulating LPS levels, changes that track with histological NAFLD improvement [151].

**Gut-Liver–Pancreas Interplay:** It is worth noting that the gut–liver and gut–pancreas axes converge in several key areas. For example, bile acids (gut–liver) can stimulate GLP-1, which effects on the liver are likely indirect, and secretion (gut–pancreas axis) via activation of Takeda G-protein-coupled receptor 5 (TGR5) on intestinal L-cells [127]. TGR5 agonists can induce brown fat activity in animal models, but adult humans have very limited active BAT, so this mechanism may have minimal clinical effect. (Hence, any BAT-mediated energy expenditure increase from TGR5 agonism is likely modest in adult humans). Notably, GLP-1 receptors are not present on hepatocytes (current evidence locates GLP-1R only on hepatic stellate cells, which is controversial). Thus, any direct hepatic actions of GLP-1 would be via non-parenchymal cells or neural pathways, not hepatocytes [158,159]. Consequently, these axes are not isolated; therapeutics often modulate multiple pathways simultaneously. A prominent example is the class of GLP-1 receptor agonists, which resolve NASH in a significant proportion of patients—likely through combined weight-reduction and anti-inflammatory actions—thereby linking gut–pancreas incretin signaling with gut–liver metabolic benefits [160].

**Table 2 nutrients-17-02708-t002:** Gut–liver-axis-targeted therapies in T2D and NAFLD.

Intervention	Molecular Target/Mechanism	Key Clinical Findings Study Population, Duration, and Outcomes	Effects on T2D and/or NAFLD
FXR Agonists, e.g., Obeticholic Acid (OCA)	Activated farnesoid X receptor (FXR) in ileum and liver, inducing FGF19 (ileal hormone) and suppressing CYP7A1 to reduce bile acid synthesis. Promoted hepatic fatty acid oxidation and insulin sensitivity while reducing lipogenesis and inflammation (FXR–FGF19 pathway).	**FLINT (Phase II)**—72 weeks in NASH (≈50% with T2D): OCA (25 mg) improved NAFLD activity score (steatosis and inflammation) and fibrosis vs. placebo. **REGENERATE (Phase III)**—18 months in NASH F1–3: fibrosis improvement ≥ 1 stage in 23% OCA vs. 12% placebo (interim analysis). Common side effects: pruritus, increased LDL cholesterol.	Improved liver histology (reduced steatosis and fibrosis) and lowered ALT/AST. In patients with T2D, FXR activation increased insulin sensitivity (with modest HbA1c reduction). Raised LDL levels and may have caused pruritus.
TGR5 Agonists, e.g., INT-777 (experimental)	Activated G-protein-coupled bile acid receptor TGR5 on enteroendocrine and immune cells. Stimulated GLP-1 and PYY release from L-cells (TGR5–GLP-1 axis) to enhance insulin secretion and satiety; increased energy expenditure in brown adipose and muscle tissues; and exerted anti-inflammatory effects via macrophage TGR5.	**Preclinical studies**: INT-777 (TGR5 agonist) improved insulin sensitivity and reduced hepatic steatosis in obese mice. A novel agonist (RDX8940) increased GLP-1/PYY and decreased liver fat in diet-induced NAFLD mice. INT-767 (dual FXR/TGR5 agonist) reduced liver fibrosis and inflammation in NASH models. **Clinical data**: No TGR5-specific agonist has been approved yet; development has been limited by TGR5-mediated gallbladder effects.	Anticipated to improve glycemic control (via incretin release) and reduce NAFLD activity (less steatosis, inflammation, and fibrosis) based on animal models. Human trials are in early phases; efficacy in T2D/NAFLD remains to be confirmed.
Probiotics Live microbiome therapy	Modulated gut microbiota composition in favor of beneficial bacteria (Lactobacillus, Bifidobacterium, etc.), strengthening the intestinal barrier and reducing endotoxin (LPS) translocation and TLR4 activation. Produced metabolites (e.g., SCFAs) that improved the host’s metabolism and reduced inflammation. May decrease microbial production of hepatotoxins (LPS, ethanol, and TMA) in the gut.	**Meta-analysis (2023 [161], 41 RCTs)**—Probiotic or synbiotic supplements significantly reduced liver fat (improved ultrasound-detected steatosis), lowered ALT, AST, and GGT, and even improved fibrosis markers in NAFLD. **RCT examples**: ~6–12-month probiotic regimens in NAFLD have shown decreased ALT/AST and improved insulin resistance (HOMA-IR) compared to a placebo. Generally well-tolerated.	Lowered liver enzymes and liver fat content in NAFLD. Modest improvements in insulin sensitivity and fasting glucose observed in T2D/MetS patients (via reduced systemic inflammation and enhanced GLP-1). Some studies reported reduced inflammatory cytokines, though effects on lipid profiles were minimal.
Bile Acid Sequestrants, e.g., Colesevelam, Cholestyramine	Non-absorbed resins that bind intestinal bile acids, interrupting enterohepatic circulation. Lower FXR activation in the ileum (disinhibiting CYP7A1), which increases conversion of cholesterol to bile acids and fecal BA excretion. Resultant effects include lowered LDL cholesterol and potentially more bile acids reaching the colon to activate TGR5 (enhancing GLP-1 release). Also, it can alter gut microbiota—reducing LPS-producing bacteria and intestinal permeability.	**T2D trials**: Colesevelam (add-on in T2D) for 12–26 weeks lowered HbA1c by ~0.5% and fasting glucose vs. placebo, and significantly reduced LDL-C. **NAFLD evidence**: A Japanese study reported improved liver enzymes and hepatic fat on imaging with colesevelam in NASH patients. However, a placebo-controlled MRI-PDFF study found no significant histological benefit (and even an increase in liver fat in the colesevelam group). A combination of an ASBT inhibitor (elobixibat) with cholestyramine is under investigation for synergistic effects.	Improved glycemic control modestly in T2D (lowered HbA1c and improved hepatic insulin sensitivity). Primarily used to reduce LDL cholesterol. In NAFLD/NASH: may modestly lower ALT and steatosis in some patients, but results are inconsistent. Has shown anti-inflammatory and anti-fibrotic effects in experimental models (via reducing gut LPS signals).
Dietary Fiber (Prebiotics), e.g., Inulin, Fructooligosaccharides	Fermentable fibers that serve as substrates for beneficial gut bacteria, leading to the production of short-chain fatty acids (SCFAs: butyrate, propionate, and acetate). SCFAs enhance gut hormone release (GLP-1 and PYY via FFAR receptors) and provide energy to enterocytes, strengthening the gut barrier. This reduces endotoxemia (LPS leakage) and liver inflammation. Also, fiber fermentation shifts microbiome composition (e.g., increases Bifidobacteria) and can reduce microbial choline conversion to TMA, mitigating fatty liver.	**RCTs in NAFLD**: Supplementation with inulin-type fructans (10–20 g/day) for 8–24 weeks (often alongside diet control) significantly reduced serum ALT and AST and lowered fasting insulin levels compared to a placebo. One trial (2020) [162] in NAFLD patients on a low-calorie diet found that adding inulin led to greater ALT reduction than diet alone. However, a recent RCT (2024) [163] with 16 g/day of inulin (and no weight loss regimen) showed improved gut Bifidobacteria but no significant change in liver fat or inflammation markers over 12 weeks.	In NAFLD, increased fiber intake was associated with reduced liver fat and aminotransferases, partly via increased SCFAs and improved insulin sensitivity. SCFAs (especially butyrate) from fiber have anti-inflammatory and insulin-sensitizing effects, aiding glycemic control. Some patients saw improved HOMA-IR and slight HbA1c reductions with fermentable fiber supplementation. Overall, dietary fiber supported weight management and metabolic health, which benefited both T2D and NAFLD.
Fecal Microbiota Transplant Microbiome transfer (FMT)	Infusion of a healthy donor’s gut microbiota to re-colonize the patient’s intestine. Aims to restore microbial diversity and beneficial commensals, leading to improved bile acid composition and SCFA production, strengthened gut barrier, and reduced production of harmful microbial metabolites (LPS and ethanol). By resetting dysbiosis, FMT targets multiple gut–liver axis pathways simultaneously.	**Metabolic syndrome (proof-of-concept)**: FMT from lean donors improved peripheral insulin sensitivity in obese subjects with metabolic syndrome within 6 weeks. **NAFLD RCTs**: Short-term FMT in NAFLD showed mixed results—one pilot RCT (21 patients) noted enhanced gut barrier function (decreased intestinal permeability) but no change in liver fat or IR at 6 weeks (Craven et al., 2020 [164]). Another RCT using a lean-vegan-donor FMT reported improved liver inflammation (histological NAS score) and shifts in hepatic inflammatory gene expression vs. autologous transplant. A larger 2022 RCT (75 NAFLD patients) found that FMT safely attenuated fatty liver and aided microbiota “reconstruction”, with improvements in liver fat and enzymes over 12 weeks.	Improved insulin sensitivity (↑ glucose uptake) in T2D/metabolic syndrome recipients following healthy FMT. Potential to reduce hepatic steatosis and inflammation in NAFLD by decreasing endotoxemia and proinflammatory signals. Some patients showed reductions in ALT and liver inflammation after FMT, though effects were variable. Long-term benefits and safety (e.g., durability of microbiome changes) are still under study.
TLR4 Inhibitors, e.g., JKB-121 (TLR4 antagonist)	Blockade of toll-like receptor 4 on Kupffer cells and other immune cells, preventing activation by LPS (an endotoxin) from the gut. Inhibiting the LPS–TLR4 pathway reduces NF-κB–mediated inflammatory cytokine release and curtails downstream stellate cell activation and fibrogenesis in the liver. The goal is to interrupt gut-derived inflammation, which drives NASH.	**Phase II trial (2018 [165])**: JKB-121 (oral TLR4 antagonist) in NASH patients (24 weeks) was well-tolerated but showed no significant improvement over placebo in liver fat, ALT, or fibrosis markers. A high placebo response rate was observed, and JKB-121 did not further reduce liver inflammation or steatosis compared to the placebo. **Other approaches**: A bovine-derived anti-LPS antibody (IMM-124/ASX-100) and gut-selective antibiotics (rifaximin) have been explored to lower endotoxemia; however, clinical efficacy in NASH remains unproven.	Preclinical models demonstrated that TLR4 inhibition can attenuate NASH progression (less inflammation and fibrosis and even reduced HCC development). However, in clinical NASH, direct TLR4 blockade has not yet improved outcomes. No meaningful effect on glycemic control or liver histology was seen with JKB-121. Targeting LPS–TLR4 remains challenging, and combination strategies may be needed for T2D/NAFLD patients.

## 7. Gut–Kidney Axis

The gut–kidney axis is an emerging concept that describes how gut-derived factors impact renal physiology and pathology and vice versa [166]. In T2D, the kidneys play a key role in glucose homeostasis (via gluconeogenesis and glucose reabsorption) and are a major target of diabetic complications (diabetic kidney disease, DKD) [167]. Chronic kidney disease (CKD), in turn, can cause changes in the gut environment (uremic toxins accumulating in the gut and altered microbiota) [168]. Two facets particularly relevant in T2D are (1) microbiota-derived uremic toxins contributing to renal injury [169] and (2) the interplay of incretin therapies (GLP-1) and SGLT2 inhibitors in gut and kidney functions [15,170,171]. Figure 4 schematically illustrates the gut–kidney axis.

**Microbial Metabolites and Diabetic Kidney Disease:** Diabetic kidney disease, affecting roughly 30–40% of T2D patients [172], is characterized by progressive loss of glomerular filtration, proteinuria, and renal fibrosis. Systemic inflammation and oxidative stress play big roles in DKD progression, in addition to hyperglycemia and hypertension [173]. The gut microbiota can produce several metabolites that become pathogenic when kidney function declines (since healthy kidneys excrete them). Among these uremic toxins are p-cresyl sulfate (pCS) and indoxyl sulfate (IS), which arise from bacterial fermentation of tyrosine and tryptophan, respectively, in the colon [174]. Normally excreted in urine, these toxins accumulate in CKD and exert proinflammatory and pro-fibrotic effects on the kidney and cardiovascular system [174,175]. Even before advanced CKD, in early DKD, elevated circulating levels of IS and pCS can be detected and are associated with worse outcomes [175]. These toxins activate oxidative stress in proximal tubular cells and stimulate TGF-β signaling, promoting interstitial fibrosis [176].

Notably, indoxyl sulfate’s precursor, indole, is produced by gut bacteria [169]; interventions that target the intestinal microbiota have been shown to lower circulating IS concentrations [177]. For example, administering AST-120— an oral spherical activated charcoal that adsorbs indole in the gut— reduces IS levels and has been reported to attenuate CKD progression in some cohorts [178,179], although large, well-powered randomized trials have yet to provide definitive benefits [180]. This illustrates the gut–kidney axis: Modulating gut-derived metabolite production can influence the trajectory of renal disease.

Conversely, CKD affects the gut: In moderate to advanced CKD, there is often gut dysbiosis—an overgrowth of proteolytic bacteria (which produce more toxins like ammonia, p-cresol, and indoles) and a reduction in fiber-fermenting taxa (partly driven by dietary restrictions and slow transit) [166,181]. CKD also leads to a build-up of urea in the blood that diffuses into the gut lumen; bacterial urease converts this urea to ammonia, raising the luminal pH and damaging the gut barrier [166,182]. CKD patients commonly exhibit increased gut permeability and systemic endotoxin levels that resemble those in metabolic endotoxemia [182,183]. Thus, a vicious cycle can form in which CKD-induced leakiness and dysbiosis amplify gut-derived inflammation, further aggravating renal and cardiovascular injuries [182,183]. In T2D, those with advanced DKD display distinct microbiome signatures relative to T2D patients without DKD; notably, DKD cohorts show depleted short-chain-fatty-acid producers and enriched opportunistic pathogens, paralleling higher circulating TMAO and IS concentrations [184]. These data underscore a mechanistic link between the gut microbial composition/metabolites and the progression of diabetic nephropathy.

**GLP-1 and SGLT2—Dual Organ Benefits:** The advent of SGLT2 inhibitors and GLP-1 receptor agonists—both originally aimed at glycemic control—has revealed surprising benefits in renal outcomes, likely via gut–kidney interactions.

**SGLT2 Inhibitors:** SGLT2 inhibitors lower blood glucose via renal glucose excretion. Notably, among them, only canagliflozin also inhibits SGLT1 in the gut, slowing intestinal glucose absorption; the others (e.g., empagliflozin and dapagliflozin) are selective for SGLT2 and do not affect gut glucose uptake. These drugs (e.g., empagliflozin and canagliflozin) block glucose reabsorption in the proximal tubule, causing glycosuria and lowering blood glucose independently of insulin. By doing so, they reduce renal hyperfiltration (excreting calories and sodium leads to lower glomerular pressure) and have been shown to slow DKD progression and reduce heart-failure risk. SGLT2 inhibitors are now recognized for their cardiorenal benefits. In fact, they substantially slow diabetic kidney disease progression and reduce the risk of kidney failure—so much so that drugs like dapagliflozin and canagliflozin are approved for chronic kidney disease, even in non-diabetic patients [185,186]. (Landmark trials, such as CREDENCE and DAPA-CKD, showed robust renoprotection.) How might they relate to the gut? First, SGLT2 inhibitors can affect the gut by increasing glucose delivery to the colon (since more glucose escapes absorption in the small intestine and spills into distal segments and stool). This extra substrate could alter the colonic microbiome—potentially enriching bacteria that consume glucose. Some rodent studies have reported that SGLT2 inhibitor treatment shifts the gut microbiota toward increased butyrate producers and Bacteroidetes [55,187]. Additionally, canagliflozin, which has partial SGLT1 inhibition in the gut, was found to increase GLP-1 levels (likely by allowing more glucose to stimulate L-cells in the distal gut or via SGLT1-mediated signaling) [188]. Indeed, a study in rodents showed that canagliflozin and related dual SGLT1/2 inhibitors suppressed GIP but enhanced GLP-1 secretion by delaying glucose absorption to the lower gut [188]. Thus, SGLT2 inhibitors may indirectly engage the gut–pancreas axis by boosting GLP-1, creating a crosstalk where a kidney-targeted drug improves the incretin response. This is supported by the observation that patients on SGLT2 inhibitors have elevated endogenous GLP-1 levels and improved β-cell function in some trials [189]. Moreover, an intriguing preclinical study demonstrated that dapagliflozin promoted pancreatic β-cell regeneration in diabetic mice, and this effect was mediated by gut microbiota and a tryptophan–GLP-1 pathway: Dapagliflozin altered the microbiome to increase the production of indole-3-propionate, which stimulated intestinal GLP-1, contributing to β-cell recovery [190]. This provides a mechanistic bridge among SGLT2 (kidney), gut microbes, and islets.

**GLP-1 Receptor Agonists:** Originally for glycemic control, GLP-1 RAs have well-documented cardiorenal benefits and have also shown renal benefits. The FLOW trial, a dedicated renal outcome study, found that semaglutide significantly slowed kidney function decline in T2D patients with CKD (the trial was stopped early for efficacy)—underscoring the kidney-protective role of GLP-1 RAs [191]. In the kidneys, GLP-1 receptors are expressed in proximal-tubule epithelial cells and, to a lesser extent, in glomerular structures [192]. GLP-1 RAs induce natriuresis and diuresis by inhibiting the Na^+^/H^+^ exchanger (NHE3) in proximal tubules, thereby lowering arterial and intraglomerular pressures [193]. Major outcome trials—including LEADER and SUSTAIN-6—have shown that GLP-1 RAs slow the progression of albuminuria and improve composite kidney end points in individuals with T2D [194,195], while concomitantly promoting weight loss and superior glycemic control, which further protects renal function. Beyond hemodynamic effects, GLP-1 signaling exerts direct anti-inflammatory actions on renal and vascular tissues, for instance, by attenuating macrophage infiltration and proinflammatory cytokine release [196]. There is also emerging gut crosstalk: In mice, treatment with the GLP-1 agonist liraglutide remodeled the microbiome—enriching Akkermansia and other beneficial taxa—and was associated with lower systemic inflammation [197]. We note that GLP-1 receptor expression in the kidneys is limited—conclusively found only on the afferent arterioles of the glomerulus. (Other locations, like tubular cells, remain unconfirmed.) Therefore, GLP-1’s renal effects are likely via hemodynamic changes (afferent arteriole dilation) or systemic pathways, rather than widespread direct action on renal parenchyma. Although human data remain limited, these findings suggest that GLP-1 therapy may confer renal benefits, partly by restoring a healthier gut ecosystem and reducing the generation of gut-derived uremic toxins.

**Combined Therapy:** Notably, in T2D patients at high risk, combining an SGLT2 inhibitor with a GLP-1 RA yields additive protection for both the heart and kidneys, and the 2022 ADA/EASD consensus now recommends this dual approach for many individuals with cardiorenal disease [198]. Mechanistically, the combination tackles several axes simultaneously: GLP-1 RAs curb appetite/weight gain and inflammation, whereas SGLT2is lower glomerular hyperfiltration and may secondarily raise endogenous GLP-1 levels and remodel the gut microbiota, as noted earlier. A 2024 systematic review and meta-analysis that stratified outcome trials by background SGLT2i use showed that GLP-1 RAs continued to confer significant cardiovascular and kidney benefits on top of SGLT2 inhibition—evidence of genuine additivity rather than redundancy [199]. Real-world data reinforce this synergy: A 2025 cohort study reported that dual GLP-1 RA + SGLT2i therapy reduced major adverse kidney events, acute kidney injury, end-stage kidney disease, and all-cause mortality more than SGLT2i alone over five years [200]. Beyond the gut–kidney axis, the same drug pairing appears to benefit the gut–liver axis: A 2023 review of NAFLD/MAFLD trials concluded that the combination improves steatosis and hepatic inflammation more than either class individually—SGLT2is primarily diminish hepatic fat, while GLP-1 RAs temper inflammatory signaling [201]. Together, these findings underline that gut-derived renal and hepatic pathways are tightly intertwined and that multi-target strategies can amplify clinical gain across organs.

**Gut–Kidney Biomarkers:** TMAO, introduced earlier, is relevant for kidney outcomes too. Elevated circulating TMAO levels predict faster CKD progression and higher all-cause mortality rates in longitudinal CKD cohorts [202]. In people with T2D, higher baseline TMAO levels are linked to a markedly greater risk of incident CKD, the doubling of serum creatinine, progression to ESKD, and overall mortality [203]. Because the kidneys normally excrete TMAO, declining renal function leads to metabolite accumulation, which can then exacerbate renal and cardiovascular damage—creating a feedforward loop. Dietary restriction of TMA precursors (e.g., red meat) or microbiota-targeted strategies to curb TMA production, therefore, represent plausible renoprotective approaches in diabetes. Phenylacetylglutamine (PAG), another gut-derived metabolite, has recently been associated with advanced CKD and heightened cardiovascular risk, underscoring its potential as an additional biomarker of gut–kidney crosstalk [204]. As high-resolution metabolomics matures, composite panels of such gut-derived metabolites may soon allow clinicians to stratify DKD risk and personalize interventions in T2D.

In summary, the gut–kidney axis in T2D highlights that maintaining gut health could be important for preventing diabetic nephropathy. Strategies like increasing dietary fiber (which can reduce uremic toxin production by altering microbial fermentation), using pre/probiotics, or novel adsorbents might complement standard renin–angiotensin blockers and SGLT2/GLP-1 therapies in the future management of DKD.

## 8. Shared Mechanisms and Integrative Model

While we have examined individual Gut-X axes, it is evident that many mechanistic themes overlap and integrate these pathways. T2D and metabolic syndrome are fundamentally multi-organ disorders with the gut at the center of a complex network. Here, we discuss some key shared mechanisms—including short-chain fatty acids (SCFAs), lipopolysaccharide (LPS), the Aryl Hydrocarbon Receptor (AhR), and others—that collectively impact insulin resistance, inflammation, and metabolic homeostasis across multiple organs. We then propose an integrative model (Figure 5) synthesizing how these signals intertwine to drive or ameliorate T2D. Figure 5 shows a conceptual framework, including speculative connections.

**Short-Chain Fatty Acids (SCFAs):** Produced by microbial fermentation of dietary fibers in the colon, SCFAs (acetate, propionate, and butyrate) have pleiotropic beneficial effects on the host’s metabolism [205]. SCFAs serve as energy substrates (accounting for up to 5–10% of humans’ daily energy) and as signaling molecules [206]. They bind to G-protein-coupled receptors FFAR2 (GPR43) and FFAR3 (GPR41) on enteroendocrine cells, immune cells, and adipose tissue cells [207]. Through these receptors, SCFAs can enhance GLP-1 and PYY release (especially propionate via FFAR2 on L-cells), leading to improved insulin secretion and appetite regulation [208]. SCFAs, particularly butyrate, also act on immune cells to promote anti-inflammatory responses; butyrate is an HDAC inhibitor that encourages regulatory T-cell (Treg) development in the gut, reducing inflammation [209].

Regarding short-chain fatty acids: Propionate: Absorbed largely by the liver and converted (via propionyl-CoA → succinyl-CoA) in the TCA cycle, propionate can serve as a gluconeogenic substrate. However, in humans, propionate is only a minor contributor to gluconeogenesis (unlike in ruminants, where it is a major glucose source). Acetate: The most abundant SCFA in circulation, acetate is taken up by many tissues (muscle, heart, and brain) and used as fuel by conversion to acetyl-CoA; a significant portion is also absorbed by the liver for lipogenesis or oxidation to CO_2_. These facts have been added to provide a realistic view of SCFA metabolism in humans rather than portraying SCFAs overly optimistically.

In the liver, acetate and propionate taken up can modulate gluconeogenesis and lipogenesis [100]. Propionate is gluconeogenic, but studies suggest that physiological levels mainly are consumed by the liver and may signal satiety in the brain via gut–brain neural circuits (through portal sensing) [210]. Butyrate, largely consumed by colonocytes, improves gut barrier integrity by serving as their primary fuel and by upregulating tight-junction proteins [211]. A stronger barrier prevents endotoxin leakage (LPS), indirectly reducing systemic inflammation [211]. Integrative effect: SCFAs improve insulin sensitivity in peripheral tissues as well [147]. One mechanism: Activation of FFAR2 on adipocytes was shown to inhibit insulin signaling in adipose tissue (short-term), leading to reduced fat accumulation and redirecting energy to muscle; mice without FFAR2 became obese, indicating that SCFA signals normally limit fat storage [212]. Human studies find that circulating SCFA levels correlate with leanness and better metabolic profiles [213].

A recent comprehensive review concludes that SCFAs contribute to metabolic health by enhancing GLP-1, improving gut barrier function, modulating inflammation, and even interacting with the autonomic nervous system; GLP-1 may help to maintain gut barrier function, but, critically, co-secreted GLP-2 is the key hormone for preserving intestinal barrier integrity [214,215]. (GLP-2, released alongside GLP-1, has potent trophic and barrier protective effects on the gut epithelium—more so than GLP-1 itself.). Fiber-rich diets that boost colonic SCFA production have consistently shown benefits: improved insulin sensitivity, reduced weight gain, and lower type 2 diabetes (T2D) incidence in cohort studies and clinical trials [216,217]. Thus, SCFAs are a unifying beneficial thread weaving through the gut–pancreas (incretin boost), gut–endocrine (satiety signals), gut–liver (reduced gluconeogenesis and improved lipid metabolism), and even gut–kidney axes (less inflammation means slower kidney damage) [100,214].

The text has been revised to temper this conclusion. Now it notes that SCFAs play beneficial roles (e.g., as energy sources and modulators of inflammation), but many of their purported hormonal effects (like GLP-1 release) are proposed mechanisms and not definitively demonstrated in humans.

**Lipopolysaccharide (LPS) and Inflammation:** LPS, a component of Gram-negative bacterial outer membranes, is a prototypical trigger of innate immune responses via TLR4 [218]. Chronically elevated LPS at low levels—termed as metabolic endotoxemia—is thought to cause subclinical inflammation contributing to insulin resistance [219]. LPS and other microbial products (such as peptidoglycan and flagellin) can enter the circulation when the gut barrier is compromised, or even via absorption with dietary fat, since chylomicrons can ferry LPS from the gut [220]. LPS activates TLR4 on macrophages in adipose tissue, liver (Kupffer cells), and even the hypothalamus, leading to NF-κB activation and cytokine release [221]. In adipose tissue, this local inflammation impairs insulin signaling in adipocytes and promotes lipolysis, releasing free fatty acids that further exacerbate insulin resistance in muscle and liver tissues [221]. In muscle tissue, studies have shown that low-dose LPS infusion in humans induces measurable insulin resistance within hours [222]. In the hypothalamus, LPS can trigger neuroinflammation, which perturbs leptin and insulin signaling for appetite regulation, thereby decreasing central sensitivity to these hormones and promoting weight gain [223].

Moreover, LPS triggers the NLRP3 inflammasome in multiple cell types (including pancreatic islet macrophages and β-cells), culminating in the maturation of IL-1β—a cytokine that directly impairs β-cell function and survival [58,224]. Indeed, metabolic inflammation (“metaflammation”) that accompanies obesity promotes progressive β-cell failure, and LPS is increasingly viewed as a key upstream instigator of this process [58]. T2D patients often exhibit chronically elevated levels of circulating lipopolysaccharide-binding protein (LBP) and soluble CD14, biomarkers that signal sustained low-grade endotoxemia [225,226]. Weight-loss interventions and specific dietary patterns—notably Mediterranean-style diets—consistently lower these endotoxemia markers in parallel with improvements in insulin sensitivity [227,228].

Therefore, LPS represents a shared “toxic” mediator of gut-derived inflammation affecting the pancreas (β-cell stress), liver (steatohepatitis), adipose tissue (inflammatory insulin-resistant state), muscles (TLR4-mediated insulin resistance), kidneys (endothelial dysfunction in glomeruli), and brain (hypothalamic inflammation) [57,219]. Reducing the LPS load—through the cultivation of a healthier microbiome (more Gram-positives and fewer Gram-negatives), strengthening the barrier integrity with butyrate and mucin producers (such as Akkermansia), and avoiding very high-fat diets that facilitate chylomicron-mediated LPS absorption—emerges as a unifying therapeutic goal for metabolic disease [229,230]. For instance, prebiotic fibers that raise colonic butyrate levels and Akkermansia abundance have been shown to lower circulating LPS levels and concomitantly improve insulin sensitivity and other metabolic parameters [229,230]. In our integrated model, a leaky, LPS-loaded gut is a central driver of systemic inflammation and insulin resistance, whereas a robust gut barrier confines LPS and prevents this pathological cascade [57].

**Aryl Hydrocarbon Receptor (AhR) and Tryptophan Metabolites:** The AhR is a cytosolic receptor/transcription factor traditionally known for mediating effects of environmental toxins (like dioxins). However, the AhR also binds various dietary and microbiota-derived indoles and polyphenols, functioning as a sensor of the gut environment [231]. The AhR is expressed in gut immune cells (especially Th17 and innate lymphoid cells) and in the liver among other sites [232]. When activated, the AhR can induce genes involved in xenobiotic metabolism (CYP enzymes) and modulate immune responses (e.g., interleukin-22 from ILC3 cells, which helps to maintain gut barrier integrity) [232]. Mounting evidence suggests that microbiota-derived AhR ligands have metabolic impacts [233]. For example, indole-3-propionic acid (IPA) from gut bacteria (mentioned earlier) activates the AhR and has been linked to improved insulin secretion and lower T2D risk [233,234]. AhR activation in the gut epithelium by commensal metabolites induces IL-22 levels, which strengthens mucosal barrier function and limits systemic inflammation—which is beneficial for metabolic health [232,235]. A deficiency of AhR ligands (due to dysbiosis or a low-fiber diet) can result in a weaker gut barrier and heightened inflammation, as seen in both inflammatory bowel disease and, possibly, metabolic syndrome [235].

On the other hand, AhR activation in the liver has complex effects: Some studies show that AhR activation can worsen steatosis by altering lipid-metabolism gene expression [236], whereas others indicate certain AhR ligands (like those from cruciferous-vegetable digestion) improve steatosis by inducing fatty acid oxidation [237]. The outcome likely depends on which ligand (different ligands cause different AhR conformations and target genes) [238]. For metabolism, balancing AhR activity appears to be the key—neither too low (which would impair gut–barrier immunity) nor too high (which could drive xenobiotic stress or pathological gene expression). Some gut bacteria (e.g., Clostridium sporogenes) produce AhR ligands, like indole-3-aldehyde, that protect against metabolic inflammation in mice [239]. In obesity models, administering an AhR-linked signal that boosts IL-22 improved gut barrier function and reduced weight gain and insulin resistance [240]. This places the AhR at a potential therapeutic node: Probiotics or diets that increase beneficial AhR ligands might combat metabolic syndrome.

**Other Shared Mediators:** A few additional points in the integrative picture:

**Branched-Chain Amino Acids (BCAAs):** As noted, BCAAs (leucine, isoleucine, and valine) are consistently elevated in insulin-resistant states and prospectively predict T2D development [241]. Gut microbiota composition markedly influences the circulating BCAA pool, with specific commensals either producing or degrading BCAAs [7]. Mechanistically, excess BCAAs chronically activate the mTOR/S6K axis in skeletal muscle, leading to inhibitory phosphorylation of IRS-1 and impaired insulin signaling [33]. Interventions that lower systemic BCAA exposure—whether by dietary restriction, pharmacological activation of BCAA catabolism, or microbiome-targeted strategies—have been shown to enhance whole-body insulin sensitivity in both rodents and early-phase human trials [242]. Finally, BCAAs intersect with other gut-derived mediators: Multi-target dietary approaches that simultaneously dampen metabolic endotoxemia (lower plasma LPS levels) and reduce BCAAs demonstrate parallel improvements in insulin resistance, underscoring crosstalk between LPS and BCAA metabolism [243].

**Endocannabinoids:** The gut microbiome can influence the host’s endocannabinoid system, which regulates appetite, pain, and inflammation [244]. Some SCFAs increase endocannabinoid levels, which promotes gut barrier function [245]. But dysbiosis can lead to elevated colonic endocannabinoid levels, which cause hyperphagia and obesity [246]. While not extensively discussed earlier, endocannabinoids link gut signals to adipose tissue and the brain [247].

**Gut Hormone Crosstalk:** We talked individually about GLP-1, PYY, ghrelin, etc. [248]. These hormones together create a postprandial “concert”. In obesity and T2D, often there are blunted PYY, higher ghrelin, and, possibly, lower cholecystokinin (CCK) responses—all tilting the energy balance toward weight gain [249,250]. So, multiple hormones are dysregulated, and each axis influences the others (e.g., bile acids can increase GLP-1 [127]; GLP-1 can indirectly affect gastric ghrelin by slowing emptying [251]).

**Inflammation and Insulin Resistance:** Nearly every shared mediator ultimately converges on inflammation or its absence. SCFAs, certain indoles, omega-3 fatty acids (also partly microbial processed) are anti-inflammatory—promoting M2 macrophages and Tregs and dampening TNFα [252,253,254]. LPS, certain saturated fatty acids (microbiota can modulate their absorption), and possibly some secondary bile acids are proinflammatory [255,256]. The balance of pro- vs. anti-inflammatory signals from the gut will determine the degree of chronic systemic inflammation, which is the final common pathway driving insulin resistance in the liver, muscles, and adipose tissues and impairing insulin secretion in the pancreas [255,256].

Taking all of this together, we can envisage an Integrative Model (Figure 5) for T2D pathogenesis and treatment opportunities.

In essence, T2D develops when harmful gut-derived signals outweigh the protective ones, leading to a self-perpetuating cycle of metabolic dysfunction. Breaking this cycle requires multi-target interventions, many of which revolve around restoring a healthy gut environment and its communication with organs. This integrative perspective highlights why combining lifestyle (dietary fiber and weight loss), microbiome-targeted therapies, and medications yields the best outcomes—they tackle different facets of the network.

Dietary fiber is fermented by gut microbiota to produce SCFAs, which activate intestinal free fatty acid FFAR2, thereby inhibiting fat synthesis, promoting satiety signals, and regulating hepatic gluconeogenesis and lipid metabolism. Butyrate, a major energy source for the colonic epithelium, enhances mucosal barrier integrity. Butyrate inhibits HDAC activity, thereby promoting the proliferation of Treg cells and exerting anti-inflammatory effects to reduce renal inflammatory responses. SCFAs stimulate L-cells to secrete GLP-1 and PYY, which act on the pancreas, brain, and gastrointestinal tract to improve islet function and energy balance.

The AhR, a key environmental sensor in the intestine, is promoted by dietary fiber. AhR activation in ILC3 and Th17 induces IL-22 secretion, strengthening the intestinal mucosal barrier’s integrity. The AhR’s role in the liver is complex: Ligand II activation may exacerbate steatosis and inflammation, while ligand I induces fatty acid oxidation and improves fatty liver.

High-fat diets promote the proliferation of Gram-negative bacteria and enhance LPS absorption, triggering systemic inflammation. LPS and NF-κB inflammatory signals induce FFA release from adipose tissue, leading to hepatic and muscular lipid deposition and insulin resistance. Hypothalamic inflammation reduces sensitivity to leptin/insulin, enhancing the appetite drive. Chronic low-grade inflammation and metabolic toxins exacerbate glomerular injury, increase IL-1β in islets, and impair insulin release. LPS induces hepatic fat accumulation, contributing to NAFLD.

Other metabolic regulatory pathways: BCAAs, associated with microbiota, activate the mTOR/S6K pathway in muscle tissue, reducing insulin sensitivity and intensifying islet metabolic stress. SCFAs regulate the eCB-signaling balance to maintain barrier function, but excessive eCB activation promotes food intake and obesity. GLP-1 and PYY, key intestinal hormones induced by gut microbiota or SCFAs, act on the pancreas and central nervous system to regulate blood glucose and appetite.

## 9. Clinical and Translational Perspectives

Appreciating the Gut-X axes in T2D opens novel clinical and translational opportunities. In recent years, there has been a paradigm shift toward therapies that modulate gut physiology or the microbiome to treat metabolic disease. Below, we discuss key perspectives: leveraging diet and microbiota interventions, developing pharmaco-microbiomics (drug–microbiome interactions), identifying microbial or metabolite biomarkers, and navigating regulatory considerations for microbiome-based therapies. The convergence of endocrinology and gastroenterology in T2D management heralds a more integrative approach to care.

**Dietary Modification and Pre/Probiotics:** Diet is the primary modulator of the gut microbiome and, by extension, Gut-X axes [257]. High-fiber diets, as repeatedly noted, increase SCFA-producing taxa and have demonstrated tangible benefits in T2D (lower HbA1c levels and weight loss) [258]. Current clinical guidelines increasingly recommend greater soluble fiber intake and fermented foods for people with T2D to support a diverse microbiota [259]. Specifically, dietary patterns, such as the Mediterranean diet or other plant-rich regimens, are linked to higher gut microbial diversity and greater production of beneficial metabolites, correlating with improved insulin sensitivity [257]. Conversely, ultra-processed, high-fat, low-fiber diets promote dysbiosis and metabolic endotoxemia, underscoring the need for nutritional counselling to correct Gut-X dysregulation [260].

Probiotics (live beneficial microbes) and prebiotics (substrates that feed good microbes) have shown modest but significant improvements in glycemic control and weight in some trials [261]. For instance, Akkermansia muciniphila, in pasteurized form, improved insulin sensitivity and lowered several cardiovascular risk markers in an obese, insulin-resistant cohort [67]. This success has fueled efforts to commercialize Akkermansia; pasteurized A. muciniphila has already received novel-food/“GRAS-equivalent” safety clearance from regulators [262]. Other probiotics, notably, specific Bifidobacterium and Lactobacillus strains, have been linked to reduced inflammatory markers and small HbA1c decreases in type 2 diabetes [261]. A persistent challenge is that efficacy varies by strain, dose, and the host’s baseline microbiome. Synbiotics (combining pro- and prebiotics) may deliver additive benefits: Co-administering inulin with a Bifidobacterium probiotic improved metabolic parameters more than either component alone in clinical studies [263].

While probiotics are widely available as supplements, their quality control is variable [264]. From a regulatory perspective, live biotherapeutic products (LBPs) intended to treat disease fall under drug-type oversight and must satisfy stringent manufacturing and clinical-trial requirements [265]. Only a handful of candidate consortia have advanced to late-stage trials: for example, a multi-strain probiotic cocktail is being tested in a randomized controlled trial for non-alcoholic steatohepatitis (NASH/NAFLD) [149], and another defined formulation (Pendulum WBF-011) has shown improved postprandial glucose control in adults with type 2 diabetes [266]. For T2D, such microbiome-based interventions are viewed as adjunctive rather than primary therapies, complementing standard pharmacological and lifestyle measures.

**Fecal Microbiota Transplantation (FMT):** FMT involves transplanting stool from a healthy donor to a patient to reset the gut ecosystem and is already an accepted therapy for refractory Clostridioides difficile infection [267]. Research is now probing its metabolic potential [268]. As noted, FMT from lean donors to individuals with metabolic syndrome significantly improved peripheral insulin sensitivity in a randomized controlled trial [269]. However, these benefits can be transient, unless recipients adopt supportive dietary changes, because the pre-existing microbiome profile may gradually re-establish [268]. FMT is also under clinical investigation for NAFLD/NASH and type 2 diabetes (T2D); initial trials report reductions in hepatic fat [152] and improvements in glycemic control for a subset of T2D participants [270]. Outcomes appear to be highly donor dependent, with a ‘super-donor’ phenomenon, whereby only a minority of donors drive most metabolic successes [268]. In practice, FMT is complex: Rigorous pathogen screening of donors is mandatory, and long-term safety for metabolic indications remains uncertain [267]. Moreover, unintended metabolic consequences have been documented—for example, significant weight gain after receiving stool from an overweight donor [271]. Thus, while promising, FMT for T2D/obesity is still experimental; if deployed, it may serve best as a metabolic ‘kickstart’ that must be consolidated by sustained diet and lifestyle modifications.

Likewise, probiotic supplements offer modest benefits (e.g., an ~0.2%HbA1c reduction) and are considered adjunctive therapies rather than definitive treatments. We have now clarified which gut-targeted interventions are proven versus which are still emerging, to avoid overstating the evidence.

**Pharmacological Modulation of Gut-X Axes:** Most current T2D drugs incidentally affect the gut. Metformin, the first-line drug, exerts a part of its action through the gut: It increases GLP-1 secretion (via microbiota reshaping and altered bile acid recirculation), improves gut-barrier function, and enriches beneficial taxa, such as Akkermansia [272,273]. Metformin also accumulates in the intestinal wall at concentrations ~30–300-fold higher than in plasma; these high gut levels and the accompanying microbiome shifts are thought to underlie its common gastrointestinal side effects [274]. To harness this gut-restricted mechanism while minimizing systemic exposure (e.g., for patients with renal impairment), delayed-release metformin preparations that dissolve mainly in the colon have been developed; randomized data show they lower glucose as effectively as conventional metformin despite minimal plasma drug levels [275].

The success of GLP-1 receptor agonists and SGLT2 inhibitors, which we have discussed from a mechanistic angle, translates to clinical practice as well—these agents are now proven to reduce cardiovascular and renal events in T2D beyond glycemic control [276,277]. Guidelines now prioritize them for T2D patients with high risk of or established cardiovascular/kidney disease [198]. Notably, these pharmacotherapies may owe a part of their efficacy to exploiting gut–organ axes (the incretin system for GLP-1 RA; caloric loss and, possibly, gut signaling for SGLT2i) [278]. The advent of dual or triple agonist drugs (e.g., tirzepatide for GIP/GLP-1 and others adding glucagon) is a direct outcome of better understanding gut hormone biology [42]. Tirzepatide showed unprecedented efficacy in weight loss (~20% in some participants) and HbA1c reduction (often in the non-diabetic range), putting many T2D patients into remission [42,89]. Interestingly, tirzepatide’s remarkable efficacy was not simply the sum of GLP-1 and GIP’s effects—other GIP/GLP-1 co-agonists did not outperform GLP-1 alone—indicating tirzepatide’s benefits were somewhat unexpected and may reflect other optimized features. These outcomes blur the line between “medical” and “surgical” managements of diabetes, as we approach the effects of gastric bypass without surgery.

On the horizon, there are other gut-targeted drugs:

Ghrelin antagonists or vaccines (to reduce hunger—though ghrelin blockers have had limited success so far) [279].

PYY analogs to enhance satiety (some in early trials, often combined with GLP-1 RA to complement one other’s appetite effects) [280].

FXR or TGR5 modulators for NASH and T2D (obeticholic acid for FXR, as we have discussed; TGR5 agonists could, in theory, increase GLP-1 levels and reduce inflammation but need to avoid gallbladder side effects, as TGR5 can relax the gallbladder sphincter) [153,281].

APE inhibitors (small molecules to tighten junctions and reduce gut permeability, an interesting concept but still early stage) [282].

Neutralizing endotoxins, e.g., bile acid sequestrants (colesevelam and sevelamer) that bind intestinal bile acids also bind LPS in the gut and was shown to lower inflammation in T2D, as well as showing modest improvements in glycemic control [283]. It is not widely used for that purpose, but it exemplifies repurposing strategies targeting gut-derived LPS.

**Biomarkers and Personalized Medicine:** One exciting area is using the gut microbiome and metabolome as a source of biomarkers to personalize T2D interventions [284]. For example, studies have shown that the baseline gut microbiota composition can predict dietary weight-loss responses—e.g., individuals with a high Prevotella-to-Bacteroides ratio lose more weight on a high-fiber diet [285,286]. In T2D, one could imagine stool or blood metabolite profiles guiding who might benefit most from a specific diet or a probiotic [284,287]. A study used microbiome data to predict postprandial glucose responses to various foods, enabling personalized nutrition plans that flattened glucose spikes [288]. Such approaches could become a part of routine diabetes management: a gut “assessment” to tailor the diet beyond generic advice [289].

For drug responses, too, gut microbes can influence drug metabolism (a field called pharmacomicrobiomics) [290]. Metformin’s efficacy, for instance, has been correlated with certain gut bacterial prevalence; if lacking, metformin might not work as well, suggesting those patients might need an alternative therapy or a concurrent probiotic to maximize the effect [291]. Another example is that microbial β-glucuronidase can reactivate certain drugs in the gut, causing side effects (as seen with a Parkinson’s disease drug, not T2D per se, but conceptually) [292]. For T2D medications, not much is known yet, but as we profile microbiomes, we might predict who responds to GLP-1 RA or SGLT2i or needs higher doses [293,294].

**Regulatory and Safety Considerations:** When introducing microbiome-modulating therapies, safety and regulation are paramount [295,296]. Probiotics are currently marketed as supplements (with minimal regulation) unless specific health claims are made; for use as a T2D treatment, far stricter quality control and robust clinical evidence will be required [295,296]. Rare cases of bacteremia caused by probiotic organisms have been reported in immunocompromised individuals, underscoring the need for careful risk stratification [297]. Fecal microbiota transplantation (FMT) is regulated as a biological product, and for indications beyond recurrent C. difficile infection, it remains experimental [298]. As companies pursue standardized oral FMT “capsules”, regulators are intensifying scrutiny of manufacturing processes and donor screening [298]. The US FDA warned in 2019 of fatal pathogen transmission after FMT in two immunosuppressed recipients, prompting calls for even tighter safety measures [299,300]. These events highlight that, even for metabolic applications, microbiome-based interventions must never compromise patient safety.

Another aspect is long-term effects: Altering the microbiome may have unexpected consequences down the line, potentially affecting immunity or raising the risk of other diseases [295,301]. For instance, expanding a single commensal could backfire: *Prevotella copri* overgrowth, when combined with a high-fiber diet, has been shown to exacerbate rheumatoid arthritis in mice, illustrating how a microbe promoted for one benefit might increase susceptibility to another disorder [302]. Ongoing monitoring and long-term follow-up in clinical trials are therefore essential, and recent regulatory guidance emphasizes systematic safety surveillance for microbiome-based products [303].

**Holistic Patient Care:** Understanding Gut-X axes encourages a more holistic approach to T2D patients [304]. It reinforces why lifestyle measures (diet and exercise) are fundamental—they do not just acutely changing glucose levels, they remodel the internal ecosystem, which dictates the disease course [305]. It also suggests multidisciplinary care: involving gastroenterologists or dietitians with expertise in the microbiome, endocrinologists focusing on obesity, etc., to co-manage patients [304]. The future may even see routine microbiome analysis in diabetes clinics, though cost and standardization need to improve [306].

In summary, the clinical translation of Gut-X axis knowledge is already underway: new therapies (incretin co-agonists and FXR agonists), microbiome modulators (diet, pro/prebiotics, and FMT), and precision nutrition approaches all stem from this understanding. Embracing these could significantly enhance T2D outcomes, addressing not just blood sugar numbers but also the underlying multi-organ dysfunction. As evidence grows, treatment guidelines are likely to incorporate more of these gut-directed strategies, moving beyond glucose-centric models to truly disease-modifying interventions.

## 10. Knowledge Gaps and Future Directions

Despite significant advances, our understanding of the gut’s role in T2D is still evolving. Key knowledge gaps remain, and addressing them will help to translate this knowledge into better therapies. Below are several pressing questions and future research directions:

**Elucidating Causality in Human Studies:** Much of our knowledge on Gut-X axes comes from animal models or associative human studies. We need more interventional studies in humans to prove causality. Future Direction: Conduct large, long-term trials of microbiome modulation (e.g., high-fiber diet, defined probiotic consortia, and FMT) in individuals at risk for T2D to see if it can prevent progression to diabetes. Such studies, akin to the Diabetes Prevention Program but focusing on gut-targeted interventions, would clarify how much altering the gut can causally reduce diabetes incidence. Incorporating multi-omics (metagenomics and metabolomics) will help to mechanistically link changes in gut composition to outcomes.

**Personalized Microbiome Therapies:** Inter-individual variability in microbiomes means a one-size probiotic may not fit all. Future Direction: Develop diagnostics to stratify patients by microbiome or metabolite profiles and then personalize interventions. For example, identify a “dysbiosis signature” that predicts poor response to metformin, and tailor alternate treatments or add a specific prebiotic for those patients. N-of-1 trials, where individual patients test different diets or probiotics with continuous glucose monitoring, could optimize personal regimens. Machine-learning algorithms can integrate diet, microbiome, and clinical data to recommend the optimal dietary composition for glycemic control.

**Mechanistic Target Discovery:** The microbiome produces thousands of metabolites; only a few (SCFAs, TMAO, indoles, etc.) have been studied in detail. Many others (e.g., bile acid derivatives, amino acid conjugates, and small peptides) could influence metabolism. Future Direction: Use untargeted metabolomics and cultured microbiome libraries to identify novel gut-derived metabolites that affect insulin sensitivity or secretion. For instance, screen microbiome metabolite libraries on pancreatic islet cells to see if any improve β-cell function or survival. Discovering new “postbiotics” (beneficial microbial metabolites) could lead to supplements or drugs that harness these effects without needing live microbes. Additionally, investigations into how exactly GIP resistance develops in T2D—is it microbiome related?—could open ways to restore GIP function.

**Gut–Brain Axis Nuances:** We know gut hormones affect appetite, but the precise neural circuits (especially in humans) are not fully charted. Also, how microbiota influence those circuits is intriguing (e.g., via vagal modulation or microbial neurotransmitters, like GABA, serotonin). Future Direction: Combine neuroimaging with microbiome interventions to observe changes in brain activity in appetite and reward centers. For example, give a probiotic or fermentable fiber supplement for 4 weeks, and use fMRI to see if the brain’s response to food cues or to gut hormone infusion changes. Understanding these connections can improve treatments for the eating behavior aspect of T2D.

**Metabolic memory:** The lasting impact of early metabolic insults despite later glycemic control may be partly mediated by gut-axis mechanisms. For instance, an initial period of dysbiosis and gut barrier leak in T2D can trigger epigenetic and immune changes that persistently sustain inflammation and metabolic dysregulation (a ‘memory’ of prior poor control). This suggests that early correction of gut dysbiosis and intestinal integrity might attenuate the metabolic memory effect, potentially reducing long-term diabetic complications.

**Microbiome and New Therapeutics:** As new medications (e.g., oral peptides and gene therapies) are developed, their interactions with gut flora are unknown. Future Direction: Research how the microbiome might metabolize or modulate next-generation diabetes drugs. For instance, if an oral peptide is inactivated by proteases from certain bacteria, co-formulation with a protease inhibitor or targeted antibiotic might be needed. Conversely, can we design drugs that specifically target microbiota to produce a metabolite in vivo (a concept of *pharmacobiotics*)? Engineering a commensal strain to secrete GLP-1 or consume excess glucose in the gut could be a futuristic therapy—there are early prototypes in mice.

**Integrated Models and AI:** The complexity of Gut-X axes (so many factors and feedback loops) means we might need computational modeling to predict outcomes of interventions. Future Direction: Develop integrative computational models (digital twins) of a patient’s metabolism that include gut microbial metabolism. These could simulate how a change (like adding a certain fiber or drug) cascades through gut microbial shifts, metabolite changes, hormonal responses, etc., to affect blood glucose. While ambitious, such models, refined by machine learning in large datasets, could greatly enhance precision medicine in T2D.

**Preventive Microbiome Strategies:** Most focus is on treating existing T2D, but can we intervene earlier? The gut microbiome is malleable in early life and influenced by diet, antibiotics, etc. Future Direction: Investigate if modulating the infant or childhood microbiome can reduce later T2D risk. For instance, children of diabetic mothers are at higher risk of obesity and T2D—do they have a distinct microbiome, and could giving prebiotics/probiotics in infancy normalize their metabolism trajectory? Longitudinal birth cohort studies linking microbiome development with insulin sensitivity outcomes would inform if early preventive measures are plausible.

**Addressing NAFLD and T2D Together:** Given the overlap of T2D and NAFLD via the gut–liver axis, future therapies might simultaneously target both. Future Direction: Test combination treatments—e.g., a GLP-1 RA plus an FMT or plus a fiber supplement—to see if together they synergistically improve both glycemic control and liver fat/inflammation. Multi-center trials could measure endpoints for both diseases. The challenge is to obtain different specialties (diabetes and hepatology) to collaborate on combined outcomes, but it makes sense for patient care, as they often have both conditions.

**Safety of Long-term Microbiome Modulation:** A gap is understanding the long-term consequences of chronically altering the gut environment. Future Direction: Establish long-term observational studies for individuals on microbiome-altering therapies (like those taking probiotics daily for years, or post-bariatric surgery patients) to monitor for any unforeseen effects, such as micronutrient malabsorption, colon cancer risk changes, or emergence of opportunistic bugs. Ensuring that boosting one beneficial microbe does not let another harmful one slip through is important.

Addressing these gaps will require interdisciplinary research—microbiologists, endocrinologists, immunologists, and computational biologists working together. The payoff will be a more complete picture of metabolic disease, moving us closer to truly curative approaches for T2D that might reprogram one’s metabolism via the gut rather than just managing blood sugar.

The future may even see routine microbiome analysis in diabetes clinics, though cost and standardization need to improve. Identifying which patients will benefit most from gut-targeted therapies is crucial—risk stratification using gut-derived biomarkers (microbial signatures or metabolites) could to help to tailor interventions to those most likely to respond. We also discuss feasibility: For instance, while gut-based therapies are promising, practical issues (e.g., ensuring sustained microbial engraftment with FMT, patient adherence to high-fiber diets or probiotics, and regulatory/safety considerations) must be addressed for success in real-world settings.

## 11. Limitations of This Review

While we have endeavored to provide a thorough synthesis, this review has several limitations. First, our literature search, though systematic, was limited to publications up to early 2025 and primarily in English. Rapidly emerging data (especially in microbiome research) may not be fully captured, and some relevant non-English studies could have been missed. There is an inherent publication bias toward positive findings in this field (studies showing links between the microbiome and T2D are more likely to be published than those finding no association), which could skew the narrative toward assuming causality or significance where the evidence is still preliminary. We attempted to cite meta-analyses and high-quality trials, but many areas rely on small sample studies or animal models, which may not generalize to broader human populations.

This review is also a narrative review, not a formal systematic review or meta-analysis. Thus, the selection of topics and papers involves subjective judgement. We chose to focus on mechanistic insights and major concepts, which means some interesting but less-developed topics (e.g., the gut–bone axis in diabetes or specific microbial species details) were omitted for brevity. Our framework of dividing into discrete axes (pancreas, liver, etc.) is somewhat artificial—in reality, these systems overlap heavily. To discuss them separately, we occasionally repeated information under different headings (like GLP-1’s effect appears in multiple sections), which could not be entirely avoided but may lead to redundancy.

To discuss each gut–organ axis separately, we had to introduce certain concepts (e.g., GLP-1 and SCFAs) in multiple sections. We have now minimized this overlap for clarity. For instance, as described in the Section 4, GLP-1’s role in insulin secretion is detailed there and only briefly noted in later sections to avoid redundancy.

Another limitation is that our discussion of clinical implications sometimes extrapolated from early research. For example, FMT and certain probiotics for T2D are not established standards of care, but we discussed them as potential strategies. The efficacy and safety of these need more evidence. We may have painted an optimistic view of microbiome therapy, whereas practical challenges (ensuring consistent microbial engraftment, patient adherence to diet changes, etc.) are significant.

We also did not deeply examine the heterogeneity of T2D—not all T2D patients have the same degree of dysbiosis or NAFLD, for instance. Patient factors, like genetics, geography, dietary habits, and medication use (e.g., many take metformin which itself alters the microbiome) can modulate Gut-X interactions. These nuances were beyond the scope of a single review and, thus, were generalized.

In terms of references, while we prioritized recent and high-impact sources, some landmark older studies were cited secondarily via reviews due to space constraints on reference list length. We might not have included all the relevant references, given the vast literature (for example, numerous individual microbial association studies are not all cited; instead, we used representative ones or reviews).

Finally, the field is fast-moving—concepts we present (like the role of specific metabolites) might be refined or even revised by future research. Thus, this review should be considered as a snapshot of the current understanding, and readers should consult the latest studies for up-to-date insights.

Despite these limitations, we aimed for a balanced and comprehensive overview. We acknowledge that further rigorous research is needed to validate many of the mechanisms and interventions discussed. We encourage readers to interpret mechanistic propositions with caution and to view this review as a framework to generate hypotheses and guide future studies rather than as a definitive textbook.

## 12. Conclusions

The paradigm of type 2 diabetes has expanded from an isolated pancreatic-centric model to a networked multi-organ disease in which the gut plays a starring role. Through a complex array of hormones, microbial metabolites, immune signals, and neural inputs, the intestine communicates with the pancreas, liver, adipose tissue, kidneys, brain, and beyond—collectively orchestrating metabolic homeostasis. When this intricate Gut-X crosstalk is disrupted by factors like dysbiosis, poor diet, and genetic predisposition, the result is a convergence of insulin resistance, impaired insulin secretion, inflammation, and weight gain that manifest as T2D and its complications.

In this narrative review, we explored the multifaceted Gut-X axes underlying T2D. We highlighted how the gut–pancreas axis (via incretins and nutrients) influences islet hormone secretion; how the gut–endocrine axis (gut–brain and gut–adipose signaling) regulates appetite and adiposity; how the gut–liver axis (through bile acids and endotoxins) contributes to NAFLD and hepatic insulin resistance; and how the gut–kidney axis (microbial toxins and hormonal crosstalk) affects diabetic kidney disease. Threaded through these axes are common mechanisms, such as SCFA-driven metabolic improvements, LPS-driven inflammation, and microbiota-modulated signaling pathways (e.g., via AhR, TGR5, or vagal circuits). An integrated model (Figure 4) illustrates that T2D is essentially a breakdown of the normal symbiotic relationship between the host and gut microbiome, leading to a cascade of pathological organ interactions.

Recognizing these connections is not merely an academic exercise—it has tangible clinical implications. Therapies targeting gut pathways (like GLP-1 receptor agonists, SGLT2 inhibitors, or even bariatric surgery) have delivered some of the most significant advances in T2D outcomes, validating the importance of Gut-X axes. Looking ahead, interventions such as personalized nutrition plans, prebiotic/probiotic supplementation, fecal microbiota transplantation, and gut-targeted pharmaceuticals hold promise to complement existing treatments and perhaps even prevent T2D in high-risk individuals. For example, modulating the gut microbiota to increase beneficial SCFAs and decrease inflammatory compounds could address root causes of insulin resistance and β-cell stress, offering a route to disease modification rather than just symptomatic control.

However, translating gut-axis science to routine therapy will require further research to fill current knowledge gaps. Large-scale trials are needed to ascertain which microbiome changes are most impactful and how to achieve them safely and consistently in diverse populations. Biomarkers derived from the gut (microbial signatures or metabolites) may soon refine risk stratification and allow tailored therapies—a step toward precision medicine in diabetes care. Moreover, a deeper understanding of gut–brain neural networks and gut–immunity interactions could yield novel targets (for instance, controlling hunger by manipulating gut hormone receptors or strengthening the gut barrier to quell inflammation).

Ultimately, the concept of “treating the gut to treat diabetes” represents a paradigm shift. It encourages collaborative care: dietitians, gastroenterologists, endocrinologists, and even microbiologists working together in managing metabolic disease. It also emphasizes preventive care through lifestyle: Diets high in fiber and low in ultra-processed foods not only help weight management but also beneficially reshape the gut ecosystem—a dual win for metabolic health.

In conclusion, T2D is not simply a disease of high blood sugar; it is a whole-body disorder arising from disordered communication among our organs and microbial partners. By viewing T2D through the prism of the Gut-X axes, we gain a more holistic understanding of its pathogenesis and identify leverage points for intervention. Continued research in this interdisciplinary arena will be crucial. With prudent application of emerging insights, we can move closer to the goal of not just controlling T2D, but **reversing it** or preventing it altogether by restoring the harmonious dialog between the gut and the organs that collectively maintain metabolic equilibrium. The ancient proverb “All disease begins in the gut” may not be entirely true in every case, but for T2D, it resonates strongly—and correspondingly, many solutions for diabetes may also begin in the gut.

## Figures and Tables

**Figure 1 nutrients-17-02708-f001:**
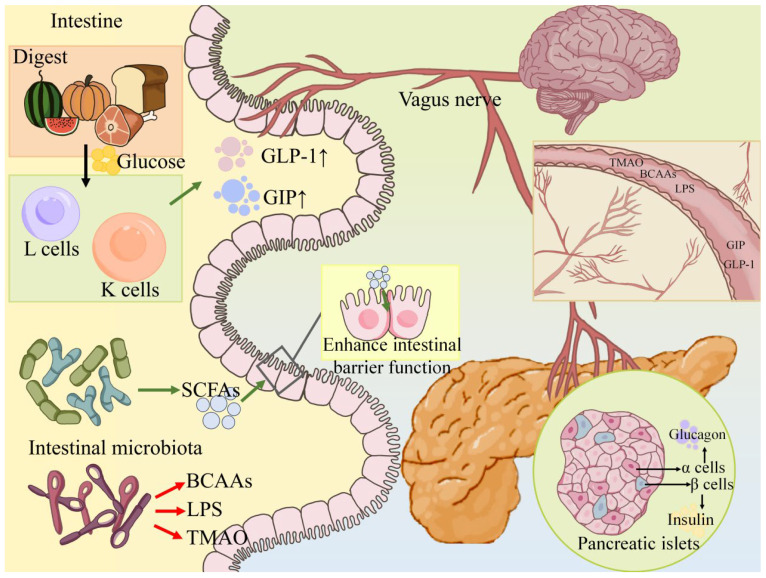
Gut–Pancreas Axis. Dietary glucose stimulates intestinal cells to secrete GLP-1 and GIP. A balanced microbiota (green microbes) produces short-chain fatty acids (SCFAs), which enhance the intestinal barrier. Conversely, a dysbiotic microbiota (red microbes) generates LPS and diabetic metabolites (e.g., TMAO and BCAAs), impairing β-cell function and insulin action. Substances such as GIP, GLP-1, BCAAs, LPS, and TMAO reach the pancreas via the vagus nerve, influencing insulin and glucagon secretion in a glucose-dependent manner. Green arrows represent protective pathways for metabolism and inflammation. Red arrows represent damaging pathways.

**Figure 2 nutrients-17-02708-f002:**
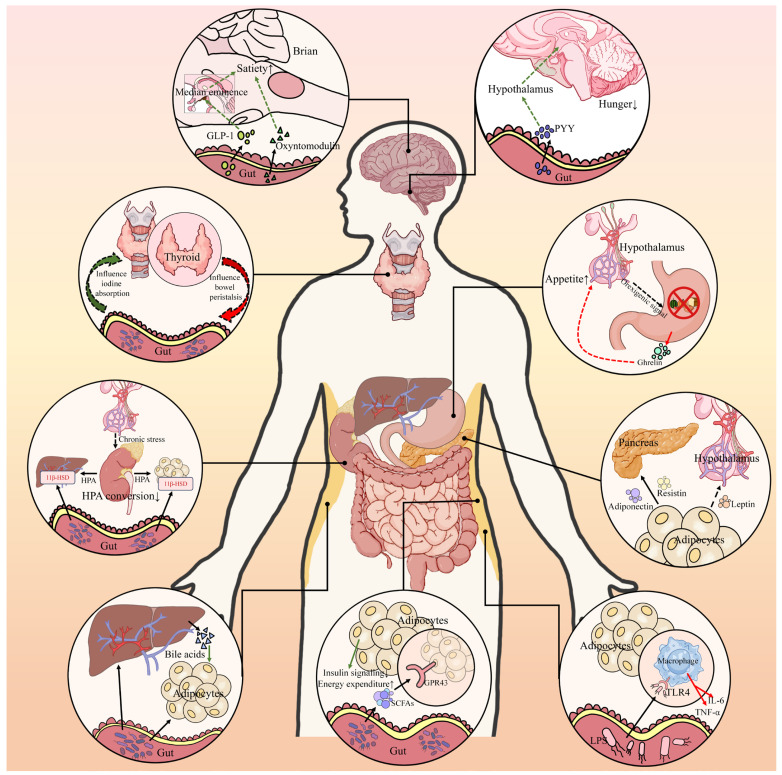
Gut–Endocrine Axis: Gut–brain signaling and interactions among the intestine, adipose tissue, and other hormone-secreting organs. The intestine and brain are intimately connected via neural and endocrine pathways. After feeding, GLP-1 and PYY is secreted by L-cells, and oxyntomodulin is secreted by the intestine, clarifying that GLP-1’s effect on the arcuate nucleus (ARC) is indirect. We note that GLP-1 acts via the median eminence (a BBB-free zone adjacent to the ARC) and possibly through tanycytes or other intermediary cells, rather than directly entering the ARC. However, physiologically, oxyntomodulin levels are low—only after bariatric surgery do its concentrations rise enough to have significant effects—so its normal role in appetite regulation remains unclear. During fasting, gastric ghrelin stimulates the same brain regions to enhance appetite. The intestine communicates with adipose tissue by regulating adipokine release and systemic lipid metabolism. Bile acids, SCFAs, and LPS have been suggested to modulate adipogenesis, energy expenditure, and inflammation (e.g., the browning of adipose tissue and activation of macrophage TLR4 pathways), though direct evidence in humans is limited and comes mainly from animal studies. Intestinal LPS infiltrates adipose tissue to activate TLR4 in adipose macrophages, inducing the production of IL-6 and TNF-α. SCFAs produced by gut microbiota bind to GPR43 receptors on adipocytes, inhibiting insulin signaling, restricting fat storage, and promoting energy expenditure. The intestine microbiota influences cortisol activity by affecting 11β-HSD in the liver and adipose tissue; the hypothalamus releases chronic stress hormones to inversely affect the kidneys’ regulation of the HPA, thereby promoting intestinal dysbiosis. The intestine–thyroid axis refers to the process where intestinal bacteria promote thyroid hormones to bind with the thyroid gland and affect iodine uptake, while hypothyroidism can, in turn, slow down intestinal peristalsis and influence the microbiota. Solid arrows denote hormone/metabolite transport via the bloodstream. Dashed arrows denote neural conduction. Green arrows represent protective metabolic and anti-inflammatory pathways. Red arrows represent damaging pathways.

**Figure 3 nutrients-17-02708-f003:**
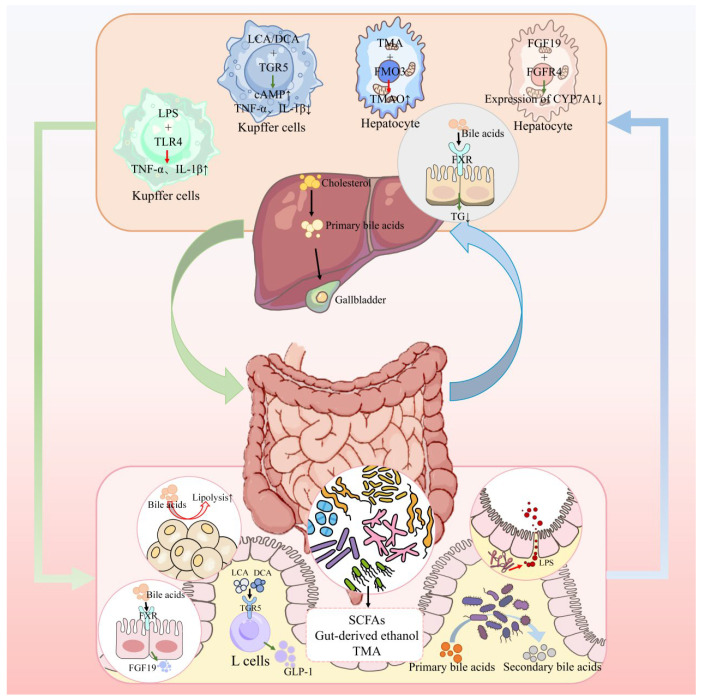
Gut–Liver Axis. Hepatocytes synthesize primary bile acids from cholesterol, which are stored in the gallbladder and released into the small intestine during meals. Bile acids activate FXR in the intestine, prompting the secretion of FGF19. FGF19 travels via the portal vein to the liver, where it inhibits CYP7A1, thereby negatively regulating de novo bile acid synthesis and reducing hepatic gluconeogenesis. FXR signaling also improves insulin sensitivity and suppresses hepatic lipid synthesis. Gut microbiota convert primary bile acids to secondary bile acids. Secondary bile acids DCA/LCA can not only activate TGR5 receptor on L-cells to promote GLP-1 secretion but also activate TGR5 receptor on Kupffer cells to increase cAMP, thereby inhibiting TNF-α/IL-1β and alleviating liver inflammation. High-fat diets induce intestinal leakage, allowing LPS to trigger the TLR4-NF-κB pathway in Kupffer cells, leading to proinflammatory cytokine release. Gut microbes produce SCFAs to regulate insulin sensitivity and gluconeogenesis; intestinal-derived ethanol exacerbates oxidative stress and lipid peroxidation, while TMA/TMAO promotes atherosclerosis and disrupts cholesterol efflux. Green arrows denote protective metabolic and inflammatory pathways; red arrows indicate damaging pathways.

**Figure 4 nutrients-17-02708-f004:**
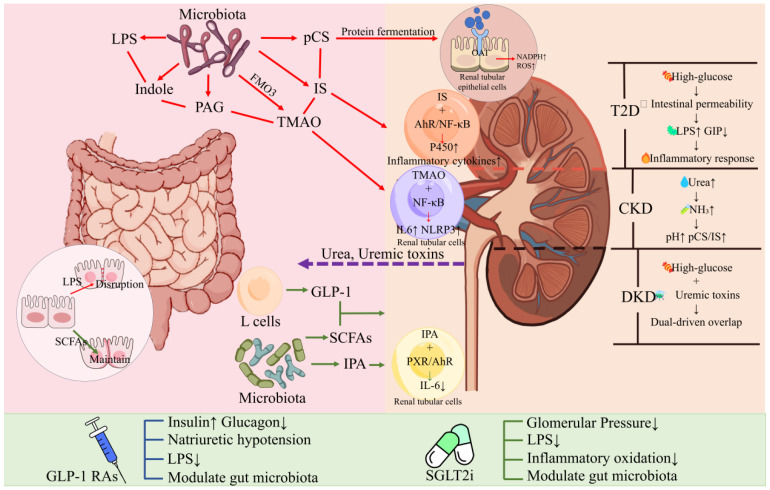
Gut–Kidney Axis: Three phases of the gut–kidney axis in type 2 diabetes (T2D), chronic kidney disease (CKD), and diabetic kidney disease (DKD). Beneficial bacteria (green microbes) produce SCFAs and IPA, while intestinal L-cells secrete GLP-1, collectively maintaining barrier function and exerting anti-inflammatory effects. IPA reduces IL-6 production by activating the PXR/AhR pathway. Harmful bacteria (red microbes) metabolize uremic toxin precursors (e.g., from p-cresol to pCS, indole to IS, TMA to TMAO, and PAA to PAG). The uremic toxin pCS enters renal tubular epithelial cells via an organic anion transporter (OAT), inducing NADPH oxidase to generate ROSs and activate inflammatory signals, such as NF-κB and NLRP3. IS exacerbates inflammation through the AhR-NF-κB-P450 pathway, while TMAO activates NF-κB to promote IL-6 and NLRP3 release. Reduced renal function allows urea and uremic toxins to diffuse into the intestine, where urease converts them to ammonia, increasing the luminal pH, disrupting the intestinal barrier, and promoting dysbiosis. Current therapeutic strategies include GLP-1 RAs and SGLT2i. Green arrows represent protective metabolic and inflammatory pathways; red arrows denote damaging pathways.

**Figure 5 nutrients-17-02708-f005:**
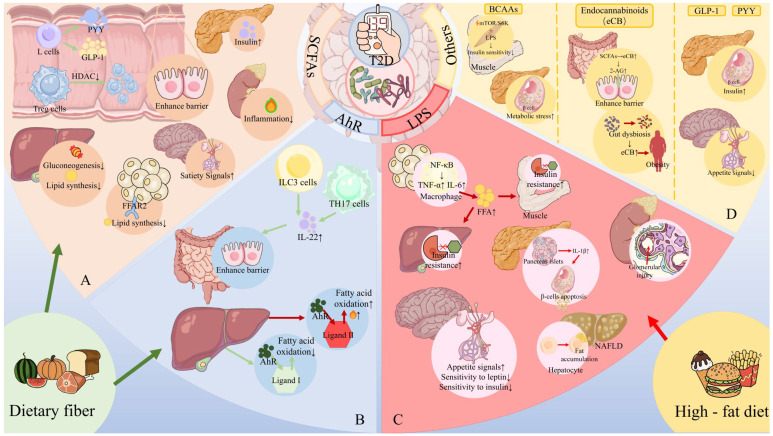
Integrative Model of Gut-X Axes in T2D. A multi-organ metabolic regulatory network centered on the gut microbiota, which systematically demonstrates that dietary fiber and high-fat diets coordinately regulate glucose–lipid metabolism, inflammatory responses, and insulin sensitivity across multiple organs by influencing microbial metabolites, immune signals, and hormonal pathways, revealing their key roles in the pathogenesis of T2D. Many pathways illustrated (e.g., microbial metabolite effects on the host’s metabolism) are hypothetical or proposed and remain to be confirmed, especially in humans. The figure shows that dietary fiber can promote intestinal microbes to produce SCFAs and AhR, while a high-fat diet promotes LPS production in the intestine. (**A**) Dietary fiber is fermented in the colon to short-chain fatty acids (SCFAs: acetate, propionate, butyrate), which act via FFAR2/3 and HDAC inhibition to stimulate L-cell secretion of GLP-1 and PYY, enhancing satiety and insulin secretion; expand Tregs and strengthen the intestinal barrier to dampen metabolic inflammation; and, via the portal vein, act on the liver to suppress gluconeogenesis and lipogenesis, improving insulin sensitivity. (**B**) Fermentation products of fiber or tryptophan metabolites activate AhR signaling, promoting ILC3/Th17-derived IL-22 to repair and fortify the barrier, thereby reducing LPS flux to the liver; hepatic fatty acid oxidation increases and lipid accumulation decreases, lowering NAFLD risk. (**C**) A high-fat diet disrupts the barrier and drives metabolic endotoxemia (LPS-TLR4), activating NF-κB and upregulating TNF-α/IL-6; overflow of free fatty acids (FFA) induces insulin resistance in liver, muscle, and adipose tissue, exacerbates β-cell stress and apoptosis, and leads to hepatic steatosis (NAFLD); central appetite and leptin signaling become dysregulated, further worsening the metabolic phenotype. (**D**) BCAAs, endocannabinoids (eCBs), and gut hormones (GLP-1, PYY) jointly regulate muscle metabolic stress, pancreatic β-cell function, and whole-body energy homeostasis; their imbalance can amplify insulin resistance and fatty liver.Green arrows signify beneficial pathways (metabolic improvement, anti-inflammation, and barrier enhancement); red arrows indicate harmful signals (proinflammation, insulin resistance, and fat accumulation).

**Table 1 nutrients-17-02708-t001:** Key components of the gut–pancreas axis in T2D.

Intervention/Mechanism	Study Design/Model	Key Findings	Relevance to T2D
GLP-1 receptor agonists (GLP-1 RAs) (incretin hormone analogs)	Multiple RCTs in T2D patients	Exogenous GLP-1 mimetics have been shown to stimulate insulin secretion and lower blood glucose in T2D, achieving ~0.9–1.0% HbA1c reduction on average (noting this depends on the baseline HbA1c) with associated weight loss. Furthermore, newer potent agents (e.g., high-dose semaglutide or tirzepatide) often produce >2.0% HbA1c declines, and over 50% of patients can attain non-diabetic HbA1c levels (e.g., >50% of tirzepatide (15 mg) patients reached <5.7%).	Established therapy leveraging the incretin effect; effective only when functional β-cells are present, demonstrating that harnessing the gut–pancreas axis can significantly improve glycemic control in T2D.
GIP hormone and dual agonists (incretin-based co-agonism)	Human physiological studies; Phase 3 trial of tirzepatide (dual GIP/GLP-1 agonist)	Endogenous GIP’s insulinotropic effect is impaired in T2D (“incretin resistance”), despite normal or high GIP levels. However, a dual GLP-1/GIP agonist (tirzepatide) produced greater HbA1c reduction and weight loss than GLP-1 therapy alone, but its superior efficacy likely reflects its higher dose rather than GIP per se—the added value of GIP remains debated.	Reveals selective incretin resistance in T2D; new dual-agonist therapies exploit the GIP pathway to further enhance insulin secretion and glycemic control.
Short-chain fatty acids (SCFAs) (butyrate and propionate—microbial metabolites)	Preclinical (gut L-cell assays and rodent models); dietary fiber RCT in T2D patients	SCFAs produced by fiber-fermenting gut bacteria engage FFAR2/3 receptors on L-cells, but whether this significantly increases GLP-1 secretion in vivo (especially in humans) is unclear. In a high-fiber diet trial, T2D patients showed expansion of butyrate-producing microbes, elevated fasting GLP-1 levels, and improved HbA1c, though the GLP-1 rise was modest, and causality remains unproven.	Suggests that altering the microbiota through diet may influence incretin pathways and improve glucose regulation in T2D, though this remains largely hypothetical.
Indole-3-propionic acid (IPA) (tryptophan-derived microbial metabolite)	Prospective cohort analysis (Finnish DPS); mechanistic in vitro studies	Higher levels of circulating IPA—a gut microbial tryptophan metabolite—were associated with lower T2D risk and better β-cell function (associative evidence). Mechanistically, IPA is hypothesized to act as an antioxidant/anti-inflammatory protector of β-cells (unconfirmed).	Identifies a beneficial microbiota-derived metabolite linked to diabetes prevention; suggests such metabolites could serve as biomarkers or therapeutic targets for preserving β-cell function.
Branched-chain amino acids (BCAAs) (microbial amino acid metabolism)	Human metabolomic studies; germ-free mouse colonization experiment	Excessive BCAA production by certain gut microbes correlates with insulin resistance and β-cell workload. Elevated BCAA levels are linked to impaired insulin sensitivity and β-cell stress. Colonizing mice with a BCAA-overproducing bacterium (Prevotella copri) worsened glucose tolerance and raised circulating BCAA levels, suggesting a causal microbiome influence.	Links gut microbiota composition (BCAA-producing species) to β-cell dysfunction and metabolic impairment in T2D; implies that modulating microbial amino acid metabolism might alleviate insulin resistance and β-cell overload.
Microbial inflammatory factors (LPS endotoxin and TMAO)	Animal models (LPS infusion and gene knockout); observational human studies	Gut-derived LPS (a bacterial endotoxin) triggers chronic low-grade inflammation via NLRP3 inflammasome activation, contributing to β-cell injury and dysfunction. Likewise, the microbial metabolite TMAO (from dietary choline metabolism) impairs insulin secretion and induces islet inflammation through NLRP3; TMAO levels are elevated in T2D and predict future diabetes. In diabetic mice, blocking TMAO production (FMO3 inhibition) restored insulin secretion and improved glycemic control.	Implicates gut dysbiosis-induced inflammation in T2D pathogenesis; interventions that reduce endotoxin leakage or TMAO formation (e.g., diet or drugs) may protect β-cells and improve metabolic outcomes in T2D.
Vagal neural reflexes (parasympathetic gut–brain–islet signaling)	Physiological studies in humans and rodents (vagal stimulation/blockade)	Nutrient ingestion activates vagal afferents in the gut, provoking a “cephalic-phase” insulin release even before blood glucose rises. Gut hormones (e.g., GLP-1) can also act via vagal pathways to prime β-cells, enhancing early-phase insulin secretion after meals.	Shows that neural signals from the gut acutely augment insulin secretion; this autonomic gut–pancreas reflex is an integral part of the postprandial insulin response and could be targeted to improve early insulin release in T2D.
Sympathetic nervous system (adrenergic signals to islets)	Clinical observations (metabolic syndrome); autonomic intervention studies	Heightened sympathetic activity (common in obesity/metabolic syndrome) inhibits insulin secretion via α-adrenergic receptors on β-cells. This increased sympathetic tone can blunt β-cell response to glucose, exacerbating hyperglycemia under insulin-resistant conditions.	Highlights that autonomic imbalance in T2D (excess sympathetic drive) negatively modulates insulin release. Reducing sympathetic overactivity (via lifestyle or pharmacological means) could thus help to un-inhibit β-cells and improve glycemic control.
Bariatric surgery (Roux-en-Y gastric bypass and similar procedures)	Clinical outcome studies in T2D patients; comparative trials vs. medical therapy	Surgical rerouting of nutrients (e.g., gastric bypass) dramatically improves glycemic control and β-cell function within days post-operatively, often before significant weight loss. The altered GI anatomy enhances distal gut nutrient delivery and incretin release (the so-called “hindgut” effect, actually a mid-gut mechanism), consistently producing an invariably exaggerated GLP-1 response that contributes to T2D remission in ~50–80% of cases.	Provides clinical proof that modifying gut physiology can send powerful diabetes-remitting signals to the pancreas. Bariatric procedures leverage the gut–pancreas axis (especially incretins) to restore euglycemia, making them among the most effective interventions for T2D.
Fecal microbiota transplant (FMT) (microbiome replacement therapy)	Randomized controlled trial in humans with metabolic syndrome (to improve insulin resistance)	Transfer of stool from lean donors to obese insulin-resistant patients led to improved insulin sensitivity at 6 weeks. Notably, in a rigorous study, only ~50% of the FMT recipients showed metabolic benefit (others did not), suggesting a responder vs. non-responder phenomenon. The benefit correlated with increased butyrate-producing bacteria. However, improvements waned by 18 weeks without diet change, indicating the microbiota shift was transient. Donor microbiome factors (“super-donors”) and concurrent diet likely influence FMT success, so its benefits are not uniform.	Proof-of-concept that modulating the gut microbiome can directly influence the host’s glucose metabolism and insulin sensitivity. Suggests that sustained microbiome-targeted therapies (possibly alongside diet) could complement T2D management by enhancing the gut–pancreas functional axis.
Probiotics (e.g., A. muciniphila) (beneficial bacterial supplementation)	Meta-analysis of 30 RCTs (1800+ patients); strain-specific RCT in T2D	Pooled RCT data show probiotic supplements yield modest but significant improvements in glycemic control (∼0.2% HbA1c reduction) and insulin sensitivity (HOMA-IR). Notably, a 12-week trial of pasteurized *Akkermansia muciniphila* in overweight T2D patients improved insulin sensitivity and lowered fasting insulin levels. Mechanistic links include enhanced GLP-1 secretion observed with certain probiotics, suggesting direct gut–endocrine benefits.	Supports the idea that optimizing gut flora can aid T2D therapy. While effects are modest, probiotics (including next-generation strains, like *A*. *muciniphila*) offer a safe adjunct to improve metabolic parameters, potentially by boosting incretin release and reducing inflammation in the gut–pancreas axis.

## Data Availability

The datasets used and/or analyzed during the current study are available from the corresponding author on reasonable request. The data are not publicly available due to privacy reasons.

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
