# Peer review of "Type 2 Diabetes and the Multifaceted Gut-X Axes"

_nutrients, 2025, doi:10.3390/nu17162708_

Round 1

Reviewer 1 Report

Comments and Suggestions for Authors

The authors  review gut homeostasis and T2D pathogenesis and then discuss several “axes” : 1) Gut-Pancreas (2) Gut-Endo- Axis (a little bit unclear but defined as: enteroendocrine signals (e.g. PYY, ghrelin) for regulation of  appetite, adipose tissue, and systemic metabolism; (3) Gut-Liver (NAFLD) and hepatic insulin resistance); (4) Gut-Kidney Axis: effect of gut-derived toxins and nutrients on diabetic kidney disease and effects of incretins  and SGLT2 inhibitor therapies. In addition is discussed microbial SCFAs improving insulin sensitivity, LPS 42 driving inflammation via TLR4, and aryl hydrocarbon receptor ligands modulating immunity. Modulating the gut microbiome and its metabolites through diet, pharmaceuticals, or microbiota therapies are discussed as therapies. Gaps for translating these insights into clinical practice are discussed.

Comments: This is a very comprehensive review covering a huge area of research. It is very informative and well written. I actually enjoyed reading it.  I think it will be useful for many people in many areas. Below I have goine into some details, where I could find inaccuracies , errors, misunderstandings etc. I have been careful to do this because I think the review might be useful for many and I would hate to see too many misunderstandings promoted.

  1. 100, while it is true that bariatric surgery can lead to diabetes remission this is not by “reshaping gut-hormone profiles and microbiota composition”, but by accelerated and abnormal exposure of distal gut segments to nutrients . Also, the success of the SGLT2inhibitors is partly due to a “plummer-like” and not normal effect on renal glucose excretion .
  2. 116. I think a new section was intended here?
  3. 130: Although this is an unusual review, I question the wisdom of also reviewing reviews. All reviews are biased. By reviewing reviews there is a serious risk of “doble bias”. I am also unhappy about exclusion of literature from before 2015. A lot of important literature in this area appeared before that time. Gastrointestinal physiology was certainly strongly developed throughout the 20th century, particularly from1950 and onwards. I also question the use of the systematic review approach in the literature search – this is an overviewing narrative review, and the most important is to be sure to include the most important communications. I don’t know what to do about the deselection of earlier literature in the review, but the editors should serious consider whether this should be encouraged.  

L.179: “gut microbial dysbiosis and increased intestinal permeability in obesity/T2D lead to 1metabolic endotoxemia, which can drive insulin resistance [28]”.- The importance of this is controversial – it should not be presented as an established fact

  1. 189: typo
  2. 195: an impaired incretin release is not established in T2DM (but impaired incretin action is)

Table 1: GLP-1 Receptor agonists are not the same as GLP-1 analogs – The term was chosen to include also exendin-derived peptides which are not GLP-1 analogs (the Gila monster has its own GLP-1 molecule which is more GLP-1 like). Regarding the A1c improvements they are said to be 0.9- 1.0 %. The relevance of this obviously depends on the baseline A1c. Decreases with tirzepatide and semaglutide are usually > 2.0 %, but even more important is the number of  individuals reaching levels below 6.5 % or even lower (more than 50 % reaching <5.7 % in the case of tirzepatide 15 mg) .

Also the question is whether GIP inclusion really increases the antidiabetic activity of the co-agonists – the biggest difference is the dose! For example  7.2 mg semagluide and tirzepatide 10-15 mg have about the same activity, at least in people with diabetes.

Also the authors proport the notion that the SCFAs promote GLP-1 release via FFAR2/3 receptors – that is wrong and the cited consequences are also erroneous  

The IPA story is still not confirmed

It is true that the mechanism of bariatric surgery is often referred to as the “hindgut effect”, but actually this is a misconception – its is not related to the hindgut but to the mid-gut (the hind gut starts at mid colon)

Also the fecal microbiota transplant results are very variable – in one of the best studies there was effect in half of the transplanted individuals – what does that show?  

As noted the effects of probiotics are weak and variable  - and it is highly doubtful whether this is a viable approach .

In conclusion, regarding table 1 – it is probably useful, but it should be more cautiously phrased -  most of the approaches are overrated.

  1. 216 It is said that there isa release of GLP-1 from the coon – in fact it has never been demonstrated that GLP-1 from the colon contributes to the circulating levels – on the contrary people with total colectomy have completely normal both fasting and postprandial GLP-1 levels
  2. 225: it is a prerequisite for the effects of the GLP-1RAs on insulin secretion that there is sufficient residual beta cell function.

l.231. it is highly controversial whether artificial sweeteners stimulate GLP-1 secretion – in the most robust studies there was no such effect.

  1. 242. As already mentioned, the actual contribution by GIP to the actions of tirzepatide remains unclear – other potent GIP/GP-1 co-agonists do not have antidiabetic properties in excess of the GLP-1 part . The case for glucagon is better (probably because it also acts on the GL1 receptor )
  2. 252: the role of SCFA to stimulate GLp-1 is highly controversial (and wrong)
  3. 263-277: If the authors had written: “it has been proposed”, all of this would be acceptable. The actions of these metabolites are almost entirely hypothetical
  4. 305: the misconception regarding the hindgut hypothesis has been alluded to above.
  5. 306: The increased GLP-1 response is invariable – not just “often”
  6. 311: this ref is not the best regarding stool transplantation - the leading group is that of Max Niewdorf
  7. 349: the physiological role of oxyntomodulin is unclear - only after bariatric surgery are the levels sufficient for this peptide to have significant effects.
  8. 351: GLP-1 does not cross the Blood brain barrier – it interacts with neurons in the circumventricular organs. The most significant expression of the receptors is also found there .
  9. 357: it should be emphasized that ghrelin levels are deceased in obesity
  10. 361: RYGB does not restrict food intake! On the contrary it allows rapid unhindered passage of nutrients from the esophagus to the more distal small intestine. The decreased food intake is exclusively due to decreased appetite.
  11. 380: note that the GIP- GLP-1 studies referred to are produced in rodents – nothing is know regarding humans.
  12. 403: there is consensus that GLP-1 does not increase EE in people

Figure 2: the upper right circle indicates that PYY interacts with the anterior pituitary – where does that come from? The problem with this kind of graphic representation that it is often inaccurate.

Also, the interaction of GLP-1 in the arcuate nucleus – this is complicated and not well worked  out; it seems there may some communication between the median eminence (to which GLP-1 has immediate access) and the neighboring arcuate  - another (but less likely) hypothesis is that there is a communication from the ventricles via tanycytes.

  1. 449-451 “Bile acids, SCFAs, and intestinal leakage-derived LPS modulate adipogenesis, energy expenditure, and inflammation by promoting adipose browning and activating the TLR4-NF- 450 κB pathway in adipose macrophages, respectively” . The evidence that this occurs in humans is really weak!
  2. 469: NAFLD is now MAFLD (recognizing that truly non-alcoholic cases are very rare) – OK, mentioned later, but why not introduce it now? l. 505
  3. 532: it should be emphasized that the main reason for hepatic insulin resistance is steatosis which is very common in obesity and T2DM-. It also should be mentioned that one of the major factors regulating hepatic overproduction of glucose in T2DM is glucagon excess. On the whole, from a therapeutic standpoint glucagon is an extremely important factor in the regulation of hepatic metabolism .
  4. 579; refs 156 – 157 Any updates regarding this compound (JKB-121) ?

588: there is agreement that there are no GLP-1 receptors on hepatocytes (but , as stated possibly on stellate cells although this is also controversial)

In table 2, section about TGR5 agonists, the authors again mention activation of BAT  - but I miss a statement on the importance of this in adult humans with very limited BAT, both here and elsewhere. Otherwise, I agree with the descriptions of the other treatments in  the table.

  1. 663: there is only one of the common SGLT2inhibitors (namely canagliflozin) that also affects SGLT1 and therefore glucose absorption in the gut – the others do not have effect.

I miss a brief summary of the impressive effects of SGLT2 inhibitors on DKD in general! This is very important and has led to approval also for non-diabetic kidney disease.

  1. 680: Again, these idea that microbiota metabolites have important effects on intestinal GLP-1 production are not supported by studies in humans at least not yet.
  2. 683: the expression of GLP-1 receptors in the kidneys is again a controversial issue – the only consensus site is the afferent arterioles! The most important study on the clinical effects is the FLOW trial!! Which very convincing and specifically deals with DKD
  3. 739: the authors mention that secreted GLP-1 may maintain the gut’s barrier function - That appears to be true, but even more important is the simultaneous secretion of GLP-2 which has marked barrier protecting effects.
  4. 755: There hasn’t been much mentioning of the AhR? I see that it comes later , but should be referred to or mentioned here
  5. 759-768 Th authors return to the effects of SCFAs on GLP-1 secretion - and as previously mentioned the evidence for this in people is very weak (in fact there is a lot of evidence to the contrary). In addition, while it is true that butyrate is a fuel for colonocytes, much less can be said about proprionate, the effect of which is humans is unclear – please outline the real metabolism of proprionate in humans ( in ruminants it represents an important stimulus to gluconeogenesis but not in humans . Acetate can be metabolized in certain tissues but what is the predominant fate of it?  Rather than hailing the SCFAs as “SCFAs are a unifying beneficial thread” it would be nice to get some facts regarding their metabolism in humans.
  6. 791: Define and explain the TLR4

Regarding Fig 4: This figures contains all the more or less hypothetical , erroneous or uncertain features discussed above – it must be possible the emphasize the hypothetical and/or controversial nature of many of these factors.- Regarding the main theme: the pathogeneses of Type2 diabetes , I accept that many of these factors may influence the development, but the main mechanisms behind T2DM have not been sufficiently emphasized, namnely: a) a genetic disposition (high heritability)  impairing beta cell function and b) obesity with development of insulin resistance (ectopic fat, liver, heart, muscle)). When glucose tolerance is impaired because of b), the poor beta cell function becomes apparent and glucose intolerance develops.  The importance of this is  apparent in cases of weight loss, where as shown in the Direct trial, a 15 % weight loss led to diabetes remission in 85 % of cases.

  1. 1011: the efficacy of tirzepatide is unique and is not due to a simple combination of GIP and GLP-1 action – such co-agonists are not more effective than GLP-1 alone. The efficacy of tirzepatide was a serendipitous finding.
  2. 1026 sevelamer should have been mentioned together with colesevelam.

Reviewer 2 Report

Comments and Suggestions for Authors

The submitted manuscript presents a scholarly and comprehensive review of the multifaceted roles of the gut and its “Gut-X axes” in the pathogenesis and management of type 2 diabetes (T2D). The work is timely and addresses an area of increasing relevance as the integration of metabolic, immune, and neuroendocrine pathways in diabetes comes to the fore. The manuscript’s organization is logical, progressing from basic concepts of gut homeostasis to the roles of gut-pancreas, gut-endocrine, gut-liver, and gut-kidney axes, culminating in clinical and translational perspectives.

The principal strengths of this review lie in its breadth of literature coverage, inclusion of high-impact and up-to-date references, and the clarity with which complex mechanistic pathways are communicated. The inclusion of summary tables and well-annotated figures enhances accessibility and aids in synthesizing a substantial body of evidence. Additionally, the discussion of shared mechanisms and an integrated multi-organ model is particularly valuable in presenting an up-to-date paradigm shift in T2D pathophysiology.

There are, however, several areas where minor revision would strengthen the manuscript:

  • Several core concepts—such as short-chain fatty acids (SCFAs), GLP-1 actions, and inflammatory signaling—are discussed in multiple sections, occasionally leading to redundancy. Streamlining these overlapping areas would improve coherence and conciseness.

  • Some discussion points on microbiome-targeted interventions (e.g., probiotics, FMT) may inadvertently overstate the strength of current clinical evidence. It is recommended to more clearly differentiate between established therapies, promising adjuncts, and experimental strategies, especially where evidence is still emerging or largely preclinical.

  • The clinical implications section might benefit from further elaboration on practical considerations such as patient stratification, cost-effectiveness, and feasibility of gut-directed therapies in real-world settings.

In summary, this is an up-to-date and informative review that makes a valuable contribution to the field. With minor revisions to address the points above, it will serve as an good resource for researchers and clinicians interested in this field.

Reviewer 3 Report

Comments and Suggestions for Authors

In the narrative review, the authors  clearly elucidated gut-mediated crosstalk with the pancreas, endocrine system, liver, and kidneys in T2DM. The article is well structured and well written. The concept Gut-X axes and initial hypothesis are clearly presented. The authors used the PRISMA guideline to prepare the review. Nevertheless, I would like to suggest that the authors expand the subsection ‘Microbial inflammatory factor’ a little in order to provide a more comprehensive view of TLR recognition dysfunction and the persistence of intestinal antigens in T2DM. In addition, the authors could discuss the mechanisms of metabolic memory in diabetes mellitus in the context of Gut-X axes regulation.

Author Response

请参阅附件
